# *Insist when Know, Caution when Not Know?* UNVEILING LLMS' FACT-CHECKING BEHAVIOR AMIDST KNOWLEDGE CONFLICTS

## ABSTRACT

By augmenting large language models (LLMs) with external evidence, tool augmentation has become a prominent approach to addressing the limitations of their static parametric knowledge in fact-checking. However, the extent to which LLMs accept external evidence when it conflicts with their internal knowledge remains unclear. Moreover, do LLMs behave consistently when they parametrically KNOW versus NOT-KNOW a specific claim? In this work, we introduce the first fine-grained evaluation framework to systematically probe LLMs' fact-checking behavior under knowledge conflicts. Our experiments reveal that most LLMs resist conflicts from external evidence when confident (KNOW) but are more receptive when uncertain (NOT-KNOW). We further demonstrate that some models (e.g., Gpt-4o-mini and Llama3-8B) achieve a better balance between openness to correct information and resistance to inaccurate evidence, whereas others (e.g., Deepseek-v3 and Gemini-2.5) tend to be either overcautious or overly credulous. To address the challenge of balancing parametric and external knowledge, we propose a test-time algorithm based on explicit Jensen-Shannon Divergence computations over sampled prediction probabilities, enabling faithful arbitration between external evidence and parametric knowledge. Our method shows competitive performance against eight baselines on our constructed FactConf datasets, improving LLM-based factuality systems in knowledge conflicts.

## 1 INTRODUCTION

After pre-training on large-scale corpora, large language models (LLMs) (Brown et al., 2020; Ouyang et al., 2022) acquire rich **parametric knowledge**, including factual information (Jiang et al., 2020; Roberts et al., 2020; Zhao et al., 2023; Li et al., 2021), which has been applied to fact-checking tasks (Tang et al., 2024; Wang et al., 2024; Vykopal et al., 2024). However, this parametric knowledge is inherently static and may contain inaccuracies or become outdated over time (Luu et al., 2021; Liska et al., 2022). A prominent line of research addressed this limitation by augmenting LLMs with external context from retrieval tools, or other models (Zhang et al., 2024; Qin et al., 2024; Shi et al., 2023). Yet, externally sourced evidence can sometimes be inconsistent with a model's internal parametric knowledge (Xie et al., 2023; Shaier et al., 2024; Pan et al., 2021) — a phenomenon we term **conflicting external evidence**. This rises an important question: how do LLMs arbitrate between knowledge derived from different sources, namely parametric versus external?

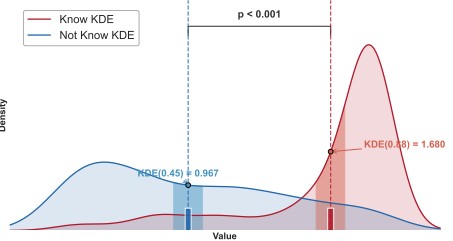

Figure 1: Probability distributions of class tokens ("True"/"False") predicted by Gpt-4o-mini based on 500 claims, with 5 runs for each one under fixed temperature 1.0. KNOW cases (red) cluster near 1, indicating high certainty, while NOT-KNOW cases (blue) show broader spread, reflecting uncertainty (Kernel Density Estimation (KDE) smoothing is applied ($p < 0.001$)).

Prior research has examined the tension between internal knowledge and external information in commonsense QA, showing that LLMs are often highly receptive to external evidence even when it

contains conflicts or inaccuracies (Xie et al., 2023; Wu et al., 2024). However, these findings may not fully extend to fact-checking scenarios, where fact verification is closely tied to the boundaries of an LLM's parametric knowledge (Ren et al., 2023; Li et al., 2024a). This leads to our central research question: *Are LLMs more inclined to reject conflicting evidence when they know the relevant facts? Conversely, are they more likely to accept conflicting evidence when they do not know the facts?*

To estimate the observable knowledge boundaries of LLMs [1], several studies quantify output uncertainty as a basis for differentiation (Li et al., 2025a; Sun et al., 2025; Chen & Mueller, 2024). Following consistency-based methods (Chen & Mueller, 2024), which define uncertainty as model's disagreement among multiple sampled predictions, we experiment by prompting an LLM (e.g., Gpt-4o-mini) for binary (True/False) claim verification to measure their internal confidence (Taubenfeld et al., 2025; Fu et al., 2025). Our results show that consistent judgments result in a sharp, low-entropy probability distribution of predicted class tokens, as illustrated in Figure 1. Based on this observation, we introduce the following categorization: claims for which the model produces identical responses across $M$ independent runs are categorized as **KNOW**, indicating confident knowledge, while all other claims are categorized as **NOT-KNOW**, reflecting uncertainty[2].

With this distinction, we conduct a fine-grained evaluation of LLMs' arbitration behavior when models encounter knowledge conflicts. Using the proposed KNOW/NOT-KNOW taxonomy, we assess seven prominent LLMs to determine whether they tend to resist or accept conflicting external evidence when they KNOW and do NOT-KNOW the facts about given claims. To support this investigation, we construct *FactConf*, a dataset for fact-checking under knowledge conflict, by automatically injecting conflicting evidence into three established fact-checking benchmarks: Quantemp, PolitiFact, and Snopes (Popat et al., 2018; Anand et al., 2024; Popat et al., 2017). In addition, we evaluate this arbitration behavior in the context of outdated knowledge, defined as factual events that occurred after a model's training cutoff and therefore lie outside its parametric memory. For this purpose, we curate *NewFactConf*, a dataset of events in 2024-25 with corresponding evidence sourced from Wikipedia.

Our fine-grained evaluation yields several notable findings:

- LLMs are more influenced in the NOT-KNOW condition than in the KNOW condition. When they KNOW, models tend to insist on their initial response, whereas when they do NOT-KNOW, they are more likely to hedge or caution against relying solely on parametric knowledge.

- Not all LLMs are naturally well-suited for handling knowledge conflicts. Specifically, Deepseek-v3, Gemini-2.5-flash, and Phi-4 frequently insist on incorrect parametric answers and refuse to revise even when provided with correct external evidence, showing lower suitability for retrieval-augmented fact-checking. By contrast, Gpt-4o-mini and Llama3-8B demonstrate greater robustness and are naturally better suited for such task.

Following these findings, we aim to help LLMs make more reliable decisions between using their own parametric knowledge or external evidence. Previous methods typically select answers based on the model's prediction confidence or the estimated reliability of external evidence (Huang et al., 2024; Bi et al., 2025). However, these approaches ignore how much model's predictions may vary across multiple runs, especially when the model is uncertain or does not know the answer, which can lead to unstable decisions. To address this gap, we provide a new approach to arbitrate between parametric knowledge and external evidence, by evaluating the consistency of model's predictions across repeated runs using a Jensen-Shannon divergence (JSD)-based metric.

Overall, our contributions are threefold:

- We introduce the first fine-grained and comprehensive evaluation framework for knowledge conflicts in fact-checking, yielding several interesting findings.

- We construct a knowledge-conflict dataset based on three public fact-checking benchmarks, and an additional benchmark requiring knowledge beyond LLM training cutoffs, testing how models handle information beyond their parametric knowledge.

---

[1] We use the term "knowledge" neutrally to refer to information stored in a model or context, explicitly noting that it is not a guarantee of truth or factual accuracy.

[2] Our categorization is agnostic to the truthfulness of a claim; that is, consistently incorrect responses are classified as KNOW. Our datasets are released at `https://doi.org/10.5281/zenodo.17197115`

- We propose an effective test-time method for making faithful arbitration decisions between the model's parametric memory and external evidence, achieving competitive fact-checking performance and stability against eight existing baselines across six LLMs.

## 2 RELATED WORKS

**Knowledge Conflict of Tool-augmented Methods**    LLMs demonstrate impressive capabilities in encoding amounts of information and supporting knowledge-intensive tasks (Petroni et al., 2021; Yang et al., 2024; Yu et al., 2023; Sun et al., 2025), including fact-checking (Zhang & Gao, 2023; Tang et al., 2024; Wang et al., 2024; Ortu et al., 2024; Mahaut et al., 2024). However, their parametric memory may become outdated, contain inaccuracies (De Cao et al., 2021; Luu et al., 2021; Liska et al., 2022), or exhibit limited memorization ability (Wang et al., 2023). To address these limitations, external tools such as retrievers are used to augment language models with up-to-date information, giving rise to tool-augmented (Qin et al., 2024; Shi et al., 2023) or retrieval-augmented LLMs (Zhou et al., 2023; Borgeaud et al., 2022; Ram et al., 2023). Yet, he retrieved evidence may itself be noisy or directly conflict with an LLM's internal parametric knowledge (Xie et al., 2023; Wu et al., 2024; Huang et al., 2024; Zhang & Gao, 2024). How LLMs arbitrate between internal and external knowledge under such conflicts, particularly in fact-checking scenarios, remains underexplored, despite its importance for trustworthy deployment of tool-augmented systems.

**Knowledge Boundary**    Recent studies have introduced the concept of the LLM knowledge boundary (Yin et al., 2024; Li et al., 2025b; 2024b), which categorizes knowledge according to model performance in knowledge-intensive question answering. Understanding this boundary is essential both for ensuring reliable deployment and for assessing model capabilities. While prior work has examined tensions between parametric knowledge and external context, such as introducing incorrect evidence, constructing counterfactual knowledge, or inducing conflicts across retrieved sources (Xie et al., 2023; Wu et al., 2024; Huang et al., 2024). However, these investigations remain insufficient for fact-checking tasks, where correctness depends on strict alignment with verifiable evidence rather than plausibility alone. In such settings, knowledge boundaries are decisive, as an LLM must resist overriding correct parametric knowledge with misleading evidence while also accepting valid new information outside its training cutoff (Ren et al., 2023; Li et al., 2024a; Ding et al., 2024; Liang et al., 2024). To address this gap, we evaluate fact-checking performance under two categories, KNOW and NOT-KNOW, thereby illuminating how knowledge boundaries shape both faithfulness and accuracy in the presence of conflicting information.

## 3 EXPERIMENTAL SETUP

**Task Formulation.**    We consider a fact-checking test set $\mathcal{Q} = \{(\mathbf{c}, \mathbf{y}, \mathbf{e})\}$, where $\mathbf{c}$ is a claim to be verified, $\mathbf{y} \in \{\text{Refute}, \text{Support}\}$ is the ground-truth label, and $\mathbf{e}$ is the associated gold evidence. We define two settings: (1) Vanilla Prediction (Parametric Only): An LLM $f(\theta)$ produces a prediction $\hat{\mathbf{y}} = f_\theta(\mathbf{c})$ using only its parametric knowledge, without access to external evidence. (2) Evidence-based Prediction: The same LLM is provided with both the claim and a set of external evidence $\tilde{\mathbf{e}}$ (which conflicts with its parametric knowledge) to predict $\hat{\mathbf{y}} = f_\theta(\mathbf{c}, \tilde{\mathbf{e}})$. The goal is to evaluate model robustness by comparing predictions across these two settings, particularly in scenarios where external evidence conflicts with the model's parametric knowledge.

**Construct Conflicting External Evidence.**    For each sample, we first obtain the vanilla prediction via $\hat{\mathbf{y}} = f_\theta(\mathbf{c})$. If $\hat{\mathbf{y}} = \mathbf{y}$, then following the controlled method of Longpre et al. (2021a), we construct conflicting evidence $\tilde{\mathbf{e}}$ via entity substitution based on the ground-truth evidence $\mathbf{e}$. Specifically, we identify all key entities $s_i \in \mathbf{e}$, and for each entity, we randomly replace it with a new entity $s'_i$ of the same type, generated by the GPT-4. The conflicting evidence $\tilde{\mathbf{e}}$ is thus created by systematically substituting all entities $s_i$ in $\mathbf{e}$ with their counterfactual counterparts $s'_i$: $\tilde{\mathbf{e}} = \mathbf{e}[s_1 \rightarrow s'_1, ..., s_i \rightarrow s'_i, ...]$. If $\hat{\mathbf{y}} \neq \mathbf{y}$, we directly use the gold evidence as the conflicting evidence, i.e., $\tilde{\mathbf{e}} = \mathbf{e}$, since it inherently represents an external source that challenges the model's internal knowledge leading to the incorrect prediction.

**Data Construction.** Based on entity substitution, we construct **FactConf**, an evaluation dataset for **Fact**-checking under knowledge **Con**flict, built with conflicting external evidence. FactConf is derived from three public fact-checking datasets that provide evidence: **Quantemp**, **PolitiFact**, and **Snopes** (Anand et al., 2024; Popat et al., 2018). Together, these dataset contain 10,866 claims with ground-truth labels and gold evidence[3], to which we apply entity substitution to generate conflicting evidence. A FactConf example is shown in Table 1, with additional examples provided in Table 13.

Table 1: Example of conflicting external evidence (partial evidence segment shown for brevity).

| Claim (c) | Label (y) | Evidence (e) | Constructed Conflicting Evidence ($\bar{\mathbf{e}}$) |
|---|---|---|---|
| McDonald's and K-pop band BTS announced a meal collaboration to be released in May 2021. | True | ... May 26, 2021 — bts' collaboration with Mcdonald's includes the BTS meal, a variety of merch and four weeks of digital content featuring the band ... | ... October 12, 2019 — bts' collaboration with KFC includes the Blackpink meal, a variety of collectibles and six weeks of digital content featuring the band ... |

To analyze LLM performance on newly emerging events, we introduce another dataset, **NewFact-Conf**, with a representative example shown in Table 11. This dataset contains 3,995 factually *true* claims crawled from Wikipedia, each accompanied with gold evidence and information indicating whether the claim is *new* for specific LLMs. Here, "new" is defined as an event that occurred after the model's publication date. Concretely, we collected *new* events as claims by crawling Wikipedia's current events portal[4], focusing on entries from year 2024 and 2025. To provide verifiable support for each claim, we gathered gold evidence from the corresponding Wikipedia references associated with each event. We further determined whether a claim should be considered "new" or "seen" for each LLM by applying a temporal cutoff based on the model's release date. Finally, we systematically created conflicting evidence for every claim using the entity-substitution protocol described in previous experiments. NewFactConf examples are provided in Appendix D.1.

While it is reasonable to assume that the LLMs did not have direct access to the relevant event-specific information in this dataset during their pre-training, one cannot be guaranteed with absolute certainty, as indirect exposure through background knowledge or data contamination remains possible. Accordingly, correct predictions on unseen events may reflect either generalization from related knowledge or hallucination, rather than explicit memorization.

**LLM Basis.** We conduct evaluation on seven leading LLMs, including both closed-source (GPT-4o-mini, Gemini-2.5) and open-source models (Mistral-7B (Mistral-7B-Instruct-v0.2), Llama3-8B, Phi4, Qwen3-32B, Deepseek-v3), spanning a wide range of scales (from 7B, 8B, 14B, 32B to 617B parameters). This diversity allows for robust comparisons across different architectures, capabilities, and development paradigms.

## 3.1 EVALUATION METRICS

**Influence Rate (IR) & Persistence Rate (PR).** The Influence Rate (IR) measures the proportion of test instances where the model's prediction changes after being exposed to conflicting evidence, reflecting its sensitivity to external context. Conversely, the Persistence Rate (PR) captures the proportion of cases where the prediction remains unchanged, indicating the model's persistence in its original prediction. Formally, these rates are defined as:

$$\text{IR} = \frac{1}{N} \sum_{i=1}^{N} \mathbb{I}\left[f_\theta(\mathbf{c}_i) \neq f_\theta(\mathbf{c}_i, \tilde{\mathbf{e}}_i)\right], \quad \text{PR} = \frac{1}{N} \sum_{i=1}^{N} \mathbb{I}\left[f_\theta(\mathbf{c}_i) = f_\theta(\mathbf{c}_i, \tilde{\mathbf{e}}_i)\right]$$

where $N$ is the total number of samples, $\mathbb{I}[\cdot]$ is the indicator function, and $\tilde{\mathbf{e}}_i$ is the conflicting evidence. By definition, PR $= 1 - $ IR, providing a complete view of the trade-off between consistency and adaptability under knowledge conflict.

---

[3]We balanced both the total volume and label distribution across three datasets, integrating them based on the size of the smallest dataset (i.e., PolitiFact, which contains 1,020 instances). As a result, the final assembled dataset consists of 3,060 samples, with half being true and the other half false.

[4]https://en.wikipedia.org/wiki/Portal:Current_events)

**Margin.** The Margin metric quantifies the difference in IR between two cases: when the model's initial prediction is incorrect and when it is correct. Intuitively, it measures how often wrong predictions are revised compared to correct ones after the model is exposed to conflicting evidence $\tilde{\mathbf{e}}_i$: Margin $= \text{IR}_w - \text{IR}_c$, where

$$\text{IR}_w = \frac{1}{|D_{\text{wrong}}|} \sum_{i \in D_{\text{wrong}}} \mathbb{I}\left[f_\theta(\mathbf{c}_i) \neq f_\theta(\mathbf{c}_i, \tilde{\mathbf{e}}_i)\right]$$

with $D_{\text{wrong}} = \{i | f_\theta(\mathbf{c}_i) \neq \mathbf{y_i}\}$ denoting the set of samples where the initial prediction is incorrect. Similarly,

$$\text{IR}_c = \frac{1}{|D_{\text{correct}}|} \sum_{i \in D_{\text{correct}}} \mathbb{I}\left[f_\theta(\mathbf{c}_i) \neq f_\theta(\mathbf{c}_i, \tilde{\mathbf{e}}_i)\right],$$

where $D_{\text{correct}} = \{i | f_\theta(\mathbf{c}_i) = \mathbf{y_i}\}$ refer to the correctly predicted samples initially. A high Margin is desirable: a low $\text{IR}_c$ indicates that the model resists changing correct answers in the face of misleading evidence, while a high $\text{IR}_w$ reflects the model's ability to revise wrong answers when provided with correct external evidence.

**Odds of Influence (OI).** We compare the odds of influence (OI) between the KNOW and NOT-KNOW states. For a given state, the odds of influence are defined as a ratio of the probability that the model's prediction changes (IR) to the probability that it remains unchanged (PR): $\text{O} = \frac{\text{IR}}{\text{PR}}$. To more precisely capture the relative difference in robustness between the two distinct states, we define the Relative Odds $\text{O}' = \frac{\text{O}_{\text{NOT-KNOW}} - \text{O}_{\text{KNOW}}}{\text{O}_{\text{KNOW}}}$, where $\text{O}_{\text{KNOW}} = \frac{\text{IR}_{\text{KNOW}}}{\text{PR}_{\text{KNOW}}}$ and $\text{O}_{\text{NOT-KNOW}} = \frac{\text{IR}_{\text{NOT-KNOW}}}{\text{PR}_{\text{NOT-KNOW}}}$. This metric quantifies how much more likely a model is to be influenced when it is in the NOT-KNOW state compared to the KNOW state. In practice, a moderately positive $\text{O}'$ is desirable: models should be more receptive to external evidence when uncertain, while still resisting misleading evidence. Very high $\text{O}'$ values, however, may indicate over-credulity, whereas negative values suggest the model is paradoxically more likely to be influenced even when it knows[5].

## 4 EVALUATION RESULTS

In this section, we present a comprehensive analysis of the experimental results to evaluate the capability of LLMs to discern and adhere to factual knowledge under different conditions. The reported results are averaged over 10 runs with a temperature of 0.3, and for the comparative evaluation presented in the paper, we used the intersection of these datasets—specifically, the smallest common set applicable to all models. We further examine the effects of varying run counts and temperature settings in Section 4.1. All the prompts are given in Appendix A.

### 4.1 MAIN RESULTS ON FACTCONF

**The distribution of KNOW and NOT-KNOW varies across LLMs.** Based on the results in Table 2, most evaluated models demonstrate high confidence in their judgments, with an overall 80% of claims falling into consistently answered categories, as reflected by the number (Ratio) between KNOW and NOT-KNOW samples. Notable exceptions are Mistral-7B and Phi-4, which exhibit significantly greater uncertainty. For Phi-4, approximately 50% of claims are classified as NOT-KNOW, indicating relatively low confidence in providing stable judgments. Mistral-7B shows even more pronounced inconsistency, with around 85% of claims producing divergent responses across multiple inference runs. This pattern suggests inherent instability and limited self-consistency in Mistral-7B for reasoning process within fact-checking contexts.

**LLMs are most influenced in the NOT-KNOW state.** As shown in Table 2, the odds of influence are generally higher in the NOT-KNOW state than in the KNOW state, indicating that models are more susceptible to conflicting external evidence when they lack prior knowledge. For instance, as shown in the correct responses, Deepseek-v3 exhibits a robustness odds of 0.687 under the KNOW condition, which rises significantly to 2.03 in the NOT-KNOW condition. A similar trend is observed when the initial answer is incorrect: Phi-4 records 0.747 under KNOW, but 1.414 under NOT-KNOW.

---

[5]This is undesirable when what it knows is correct, but can be desirable when what it knows is wrong.

Table 2: Performance on FactConf given conflicting external evidence, evaluated by influence rate (IR), persistence rate (PR), and odds of influence (O). The Ratio is the sample count ratio for Know or Not-Know under each model, separated as initial CORRECT (Left) and WRONG (Right) responses.

| | KNOW | | | | NOT-KNOW | | | | KNOW | | | | NOT-KNOW | | | |
|---|---|---|---|---|---|---|---|---|---|---|---|---|---|---|---|---|
| | IR | PR | O | Ratio | IR | PR | O | Ratio | IR | PR | O | Ratio | IR | PR | O | Ratio |
| **Deepseek-v3** | 40.75% | 59.25% | 0.687 | 60.75% | 66.99% | 33.01% | 2.030 | 3.42% | 28.30% | 71.70% | 0.395 | 32.15% | 55.95% | 44.05% | 1.271 | 3.68% |
| **Gpt-4o-mini** | 24.73% | 75.26% | 0.329 | 61.73% | 57.59% | 42.40% | 0.345 | 1.78% | 59.77% | 40.23% | 1.487 | 33.92% | 74.67% | 25.33% | 2.949 | 2.57% |
| **Qwen3-32B** | 29.75% | 70.25% | 0.423 | 84.57% | 47.37% | 52.63% | 0.900 | 3.15% | 49.41% | 50.59% | 0.977 | 10.85% | 39.18% | 60.82% | 0.644 | 1.43% |
| **Gemini-2.5** | 43.65% | 56.35% | 0.776 | 50.00% | 60.02% | 39.98% | 1.502 | 10.00% | 43.67% | 56.33% | 0.775 | 19.17% | 51.50% | 48.50% | 1.062 | 20.83% |
| **Llama3-8B** | 30.80% | 69.19% | 0.445 | 46.38% | 40.16% | 59.84% | 0.672 | 14.87% | 70.05% | 29.95% | 2.339 | 26.73% | 67.00% | 33.00% | 2.030 | 12.02% |
| **Mistral-7B** | 26.36% | 73.64% | 0.358 | 5.45% | 27.26% | 72.74% | 0.375 | 9.29% | 28.41% | 71.59% | 0.397 | 0.68% | 44.24% | 55.76% | 0.793 | 84.58% |
| **Phi-4** | 42.76% | 57.24% | 0.747 | 22.62% | 58.57% | 41.43% | 1.414 | 19.71% | 50.56% | 49.43% | 1.023 | 19.31% | 51.16% | 48.84% | 1.048 | 38.36% |

Table 3: Comparison of Margin and Relative Odds (O′) across models, showing O′ for initial CORRECT (left) and WRONG (right) responses on the FactConf dataset.

| | Gpt-4o-mini | Deepseek-v3 | Qwen3-32B | Gemini-2.5-flash | LLama3-8B | Mistral-7B | Phi-4 |
|---|---|---|---|---|---|---|---|
| Margin ↑ | +1.881 | -0.526 | +0.149 | -0.221 | +1.626 | +0.229 | -0.045 |
| Relative Odds (O′) (CORRECT) | 4.86% | 195.49% | 112.77% | 93.56% | 51.01% | 4.75% | 89.29% |
| Relative Odds (O′) (WRONG) | 98.32% | 2.22% | -34.08% | 37.03% | -13.21% | 99.75% | 2.44% |

Similarly, 5 out of 7 models have higher odds in the NOT-KNOW state for the wrong responses, while some models are not more influenced when they do not know (e.g., Qwen3-32B and Llama3-8B). Moreover, as shown in Table 3, regardless of whether the initial answer is correct or not, 12 out of 14 O′ values are positive, further confirming that LLMs are more likely to be influenced in the NOT-KNOW state than in the KNOW state. An interesting finding is that the O′ values of Deepseek-v3 and Qwen3-32B are excessively large for correct responses, indicating that they tend to be overly credulous when they do not know the answers, even though they get them right incidentally. Qwen3-32B and Llma3-8B have negative O′ when they're wrong, suggesting they are inclined to rectify the answer by accepting external evidence, which is confirmed by their positive margin.

**Not all LLMs are naturally well-suited for handling knowledge conflict scenarios.** Table 3 shows that Margin values vary widely across different LLMs, revealing substantial differences in their ability to process and reconcile conflicts with external evidence. Gpt-4o-mini, Qwen3-32B, Llama3-8B, and Mistral-7B all exhibit positive Margin, indicating greater resistance to conflicts when their initial answers are correct, while remaining more susceptible—often accepting external evidence—when their initial answers are incorrect. Larger Margins, such as those observed in Gpt-4o-mini (1.881) and Llama3-8B (1.626), suggest stronger robustness under knowledge conflict, highlighting their particular suitability for retrieval-augmented fact-checking. In contrast, models such as Deepseek-v3, Gemini-2.5-flash, and Phi-4 appear less effective for such task, as they show a stronger tendency to trust their own predictions even when those predictions are incorrect. When comparing the values in Table 2 for CORRECT and WRONG responses (left vs. right), we clearly observe this phenomenon in Deepseek-v3. It demonstrates a significant decline in robustness, with O = 0.687 for correct responses compared to 0.395 for incorrect ones. This suggests a tendency to exhibit overconfidence in erroneous answers. This variability underscores the importance of careful model selection for fact-checking task, where knowledge conflict must be evaluated and resolved.

**Influence Rate (IR) of NOT-KNOW shows limited sensitivity to the number (#) of runs.** We examine the impact of the # of independent runs (up to 40) and temperature settings (0.3, 0.5, 1.0) to the consistency of model outputs and the relationship between the IR for KNOW and NOT-KNOW cases. The Figure 3 shows the analysis of Llama3-8B, the proportion of NOT-KNOW claims stabilizes after approximately 20 runs (consistently across all three temperature settings), indicating a clear saturation point. In the middle and right panels, the IR for KNOW cases decreases steadily with more runs, reflecting growing model confidence, whereas the IR of NOT-KNOW cases shows only a negligible reduction. This confirms that the NOT-KNOW category effectively captures intrinsic model uncertainty—a state in which further consistency checks do not improve confidence. Analysis of # of runs and temperature based on other models is provided in Appendix A.

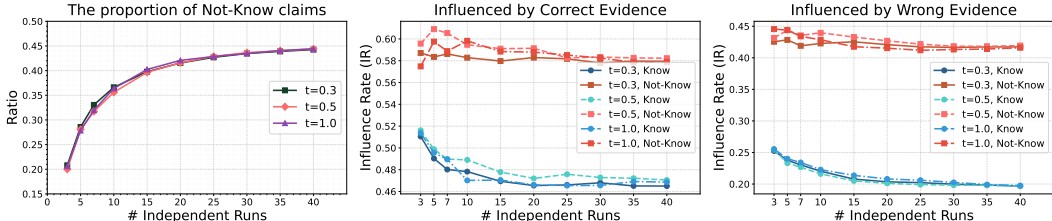

Figure 2: The analysis based on the number of independent runs. (Left) The proportion of NOT-KNOW cases plateaus after reaching a certain number of runs. (Middle & Right) The IR of KNOW cases decreases more significantly with increasing number of runs when exposed to conflicting evidence, whereas NOT-KNOW cases demonstrate relatively stable IR throughout the process.

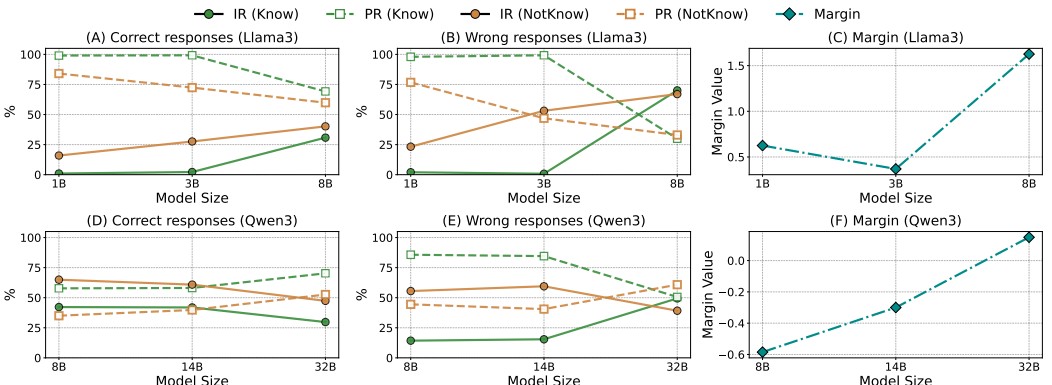

Figure 3: Model Scaling Evaluation. The influence rate (IR), persistence rate (PR), and margin results of different models in various sizes – Llama3 (1B, 3B, 8B), Qwen3 (8B, 14B, 32B).

**Model Scaling Evaluation.** We have systematically evaluated the scaling properties of the Llama3 and Qwen3 model families, revealing distinct behavioral patterns across different model sizes. As shown in Figure 3, the Margin score exhibits a clear positive correlation with model scale in both families. The Qwen3 series shows a consistent increase from –0.585 (8B) to 0.149 (32B), while the Llama3 series displays a notable performance leap at the 8B scale, reaching a Margin of 1.626. This overarching trend suggests that increasing model size enhances conflict resolution capability. Further analysis in Figure A, B and D, E shows that smaller-scale models (e.g., Llama3-1B/3B and Qwen3-8B/14B) exhibit strong response inertia, maintaining high Persistence Ratios even when incorrect. In contrast, as model scale increases, we observe a significant divergence: larger models demonstrate higher Persistence after correct answers and increased Implicit Reversal after errors. This behavioral shift indicates that model scaling improves the arbitration of knowledge conflicts and strengthens self-awareness of knowledge boundaries.

## 4.2 RESULTS ON NEWFACTCONF

Observing the results in Table 4 and Table 2, it is evident that the odds of influence are significantly higher in the NewFactConf data than in FactConf. This highlights the importance of external evidence in maintaining factual accuracy, especially considering that prior parametric knowledge may be outdated. For instance, Deepseek-v3 shows an O value of 67.97 in the NOT-KNOW state on NewFactConf, compared to just 2.030 in the NOT-KNOW state of FactConf. Similarly, Gpt-4o-mini exhibits O values of 40.44 vs 0.345, respectively. Furthermore, in the KNOW state on NewFactConf, the initial correct responses' number ratio is substantially lower than initial wrong responses, as compared to FactConf. This trend is intuitive for claims related to *new* events.

In Table 4, Phi-4 and Gemini-2.5-Flash exhibit a strong tendency to refute *new* claims—with Phi-4 supporting none of them. This reflects an overly conservation when encountering *unseen* events. Correspondingly, these models demonstrate low acceptance of correct external evidence, as indicated by their low IR (Phi-4: 24.39% and 27.45%; Gemini-2.5: 28.65% and 24.35%). Such behavior suggests that these models adopt an exceptionally cautious approach toward potentially unseen facts.

Table 4: Performance on NewFactConf given conflicting external evidence, evaluated by influence rate (IR), persistence rate (PR), and odds of influence (O). The Ratio is the count ratio over 10 runs for each model, separated as initial CORRECT (Left) and WRONG (Right) responses.

| | KNOW | | | | NOT-KNOW | | | | KNOW | | | | NOT-KNOW | | | |
|---|---|---|---|---|---|---|---|---|---|---|---|---|---|---|---|---|
| | IR | PR | O | Ratio | IR | PR | O | Ratio | IR | PR | O | Ratio | IR | PR | O | Ratio |
| **Deepseek-v3** | 88.02% | 11.98% | 7.347 | 7.35% | 98.55% | 1.45% | 67.97 | 2.64% | 46.58% | 53.42% | 0.872 | 45.82% | 68.25% | 31.75% | 2.150 | 44.19% |
| **Gpt-4o-mini** | 76.49% | 23.51% | 3.218 | 43.56% | 59.56% | 40.44% | 40.44 | 4.56% | 54.00% | 46.00% | 1.524 | 44.73% | 68.25% | 31.75% | 9.23 | 7.15% |
| **Qwen3-32B** | 75.48% | 24.52% | 3.078 | 27.72% | 88.35% | 11.65% | 7.583 | 7.86% | 90.90% | 9.10% | 9.989 | 51.14% | 79.38% | 20.62% | 3.850 | 13.28% |
| **Gemini-2.5** | 0.00% | 100.0% | 0.000 | 0.25% | 77.78% | 22.22% | 3.500 | 0.75% | 28.65% | 71.35% | 0.402 | 48.00% | 24.35% | 75.64% | 0.322 | 51.00% |
| **Llama3-8B** | 28.60% | 71.40% | 0.317 | 50.80% | 26.39% | 73.61% | 0.532 | 8.24% | 94.81% | 5.19% | 9.460 | 15.44% | 98.85% | 1.15% | 38.37 | 25.52% |
| **Mistral-7B** | 63.65% | 36.25% | 1.756 | 15.88% | 72.51% | 27.49% | 2.638 | 56.72% | 27.80% | 72.20% | 0.385 | 9.01% | 40.50% | 59.50% | 0.681 | 18.39% |
| **Phi-4** | - | - | - | 0.00% | - | - | - | 0.00% | 24.39% | 75.61% | 0.323 | 94.85% | 27.45% | 72.55% | 0.378 | 5.15% |

In contrast, Gpt-4o-mini and Qwen3-32B show considerably higher IR values across both KNOW and NOT-KNOW conditions, suggesting they are more susceptible to external evidence—whether correct or incorrect—when processing unseen events. While their higher consistency in certain cases might stem from stronger generalization and reasoning capabilities, it could also be partly attributed to hallucinatory behavior. Further analysis on this trade-off is provided in § A.5.

## 5 METHOD AND RESULTS

We focus on enabling LLMs to make more reliable choices between parametric knowledge and external evidence. Prior approaches typically compare the model's own prediction confidence (often measured through token-level probabilities) with the assessed reliability of external evidence, such as evidence correctness scores, to select the most accurate answer (Huang et al., 2024; Bi et al., 2025). However, these methods overlook variations across multiple model runs, particularly in cases where the model is in the NOT-KNOW state. This omission reduces sensitivity when deciding between parametric and external-evidence-based answers under uncertainty.

To address this limitation, we assess the stability of LLM decisions across parallel runs by computing a Jensen-Shannon divergence (JSD)-inspired measure to capture the consistency of the model's choices between parametric knowledge and external evidence, enabling more robust and informed selection. Unlike prior work, our approach emphasizes cross-run stability rather than single-instance confidence, providing a more reliable arbitration strategy in the presence of knowledge conflicts.

**Stability-based Arbitration.** Given a test set $\mathcal{Q} = (\mathbf{c}, \mathbf{y})$, where $\mathbf{c}$ is the claim to be checked and $\mathbf{y} \in \{\text{Refute, Support}\}$ is the ground-truth label, we use an LLM $f_\theta$ to sample $K$ prediction probabilities for the "True" token under a set of external evidence $\mathbf{e}$: $\{p_{\text{ext}}^{(i)} = f_\theta(\mathbf{c}, \mathbf{e})\}_{i=1}^K$.

For each sample, the binary entropy is defined as:

$$H(p_{\text{ext}}^{(i)}) = -p_{\text{ext}}^{(i)} \log_2 p_{\text{ext}}^{(i)} - (1 - p_{\text{ext}}^{(i)}) \log_2 (1 - p_{\text{ext}}^{(i)}).$$

We then compute the JSD across the $K$ samples: (1) Calculate the mean probability $\bar{p}_{\text{ext}} = \frac{1}{K} \sum_{i=1}^K p_{\text{ext}}^{(i)}$ and its entropy $H(\bar{p}_{\text{ext}})$; (2) Compute the expected individual entropy $\mathbb{E}[H(p_{\text{ext}})] = \frac{1}{K} \sum_{i=1}^K H(p_{\text{ext}}^{(i)})$; (3) The stability metric based on JSD is then computed as:

$$\mathbf{JSD}_{\text{ext}} = H(\bar{p}_{\text{ext}}) - \mathbb{E}[H(p_{\text{ext}})]$$

which corresponds to the JSD for the binary case, quantifying the uncertainty of the model's outputs given external evidence. Lower $\mathbf{JSD}_{\text{ext}}$ values indicate higher consistency across multiple runs.

Analogously, given $K$ parametirc prediction probabilities $\{p_{\text{par}}^{(i)} = f_\theta(\mathbf{c})\}_{i=1}^K$ for the "True" token without external evidence, we compute $\mathbf{JSD}_{\text{par}}$ for the parametric case:

$$\mathbf{JSD}_{\text{par}} = H(\bar{p}_{\text{par}}) - \mathbb{E}[H(p_{\text{par}})]$$

Finally, we define the difference $\Delta = \mathbf{JSD}_{\text{par}} - \mathbf{JSD}_{\text{ext}}$ and compare it against a threshold $\tau \in [-0.08, -0.06]$, calibrated on a held-out validation set (see §C), to decide whether to rely

Table 5: Accuracy (%) for correct (CE) and wrong (WE) external evidence, and the overall accuracy (OA). The best and second-best performances are highlighted in **bold** and underline, respectively.

| Baseline | Gpt-4o-mini | | | Phi-4 | | | Mistral-7B | | | |
|---|---|---|---|---|---|---|---|---|---|---|
| | WE. | CE. | OA. | WE. | CE. | OA. | WE. | CE. | OA. | Overhead |
| ImplicitSCR | 45.48 | 59.88 | 52.68 | 50.10 | 60.80 | 55.45 | 34.56 | 38.83 | 36.70 | 2 |
| ExplicitSCR | 54.12 | 64.35 | 59.24 | 41.81 | 50.84 | 46.33 | 10.33 | 18.77 | 14.55 | 2 |
| InternalEval | 51.43 | 57.60 | 54.52 | 47.60 | 48.80 | 48.20 | 6.26 | 3.08 | 4.67 | 3 |
| ContextEval | 55.20 | 67.03 | 61.12 | **52.50** | **66.60** | **59.55** | 27.61 | 10.33 | 18.97 | 2 |
| ContextConf | 47.90 | 64.70 | 56.30 | 52.20 | 52.20 | 52.20 | 15.33 | 10.33 | 12.83 | 2 |
| InternalConf | 47.60 | 48.40 | 48.00 | 46.60 | 65.20 | 55.90 | 34.33 | 34.33 | 34.33 | 2 |
| TPC | 46.90 | 61.60 | 54.25 | 51.14 | 51.14 | 51.14 | 43.33 | 57.67 | 50.50 | 2 |
| TACS-LR | 44.60 | 54.00 | 49.30 | 45.20 | 51.80 | 47.15 | 37.20 | 47.80 | 42.50 | 2 |
| Ours (K=5) | **55.38** | **67.14** | **61.26** | 52.40 | 63.60 | 58.00 | **51.20** | **62.96** | **57.08** | 5 |

| | Qwen3-32B | | | Gemini-2.5-flash | | | Llama3-8B | | | |
|---|---|---|---|---|---|---|---|---|---|---|
| | WE. | CE. | OA. | WE. | CE. | OA. | WE. | CE. | OA. | Overhead |
| ImplicitSCR | 47.77 | 60.08 | 53.93 | 14.90 | 21.85 | 18.38 | 50.40 | 60.00 | 55.20 | 2 |
| ExplicitSCR | 19.27 | 23.55 | 21.41 | **57.50** | **66.93** | **62.22** | 32.60 | 38.60 | 35.60 | 2 |
| InternalEval | 52.63 | 59.29 | 55.96 | 49.95 | 55.81 | 52.88 | 51.04 | 58.39 | 54.72 | 3 |
| ContextEval | **58.49** | 64.25 | 61.47 | 55.61 | 38.73 | 47.17 | 54.92 | 61.17 | 58.05 | 2 |
| TACS-LR | 45.20 | 51.80 | 48.50 | 39.50 | 53.00 | 46.25 | 28.60 | 34.60 | 31.60 | 2 |
| Ours (K=5) | 56.80 | **69.12** | **62.97** | 51.12 | 65.01 | 58.07 | **59.00** | **65.20** | **62.10** | 5 |

on parametric knowledge or external evidence:

$$\hat{\mathbf{y}} = \begin{cases} f_\theta(\mathbf{c}), & \text{if } \Delta \leq \tau \\ f_\theta(\mathbf{c}, \mathbf{e}), & \text{otherwise.} \end{cases}$$

Our algorithmic implementation of this stability-based arbitration method is provided in §C.

**Baselines & Implementation.** Following Huang et al. (2024), we evaluate eight methods for evidence reliability assessment under six prominent LLMs, grouped into three categories (with more details in §E): **Self-guided methods**: **1)** ImplicitSCR: The LLM is prompted with a claim and potentially incorrect evidence, then judges the reliability of the evidence before producing a final response. **2)** ExplicitSCR: The LLM generates answers using internal knowledge and external context, performs explicit confidence reasoning via CoT and answer comparison, and then balances both sources to make a final decision. **Rule-based methods**: **3)** InternalEval: The LLM evaluates its own initial parametric response; if deemed correct, it is kept, otherwise a new answer is generated using external evidence. **4)** ContextEval: The LLM evaluates external evidence; if correct, the evidence is used, otherwise it responds only with the claim. **5)** InternalConf (Jiang et al., 2023b): The LLM chooses between internal and evidence-based answers based on token-level probabilities, using a predefined threshold. **6)** ContextConf: Similar to InternalConf but applied to evidence-first responses, i.e., the selection is based on prompting with evidence first. **7)** TPC: Compares confidence scores between internal and context-based answers, using raw probabilities or calibrated percentiles. **Context-based methods**: **8)** TACS-LR: Filters unreliable context using LLM-based removal, then uses the refined context to generate the final answer Yu et al. (2024); Huang et al. (2024).

Baseline methods (based on token-level probabilities or context evaluation) and our approach, relying on LLM assessment, did not yield reliable probability estimates or judgments when applied to NewFactConf with events occurring after model's training cutoff, as demonstrated in § 4.2. We experiment exclusively on the FactConf dataset and leave methods for future events to future work.

**Metrics.** Given a test set $\mathcal{Q} = \{(\mathbf{c}, \mathbf{y})\}$, where $\mathbf{c}$ denotes a claim and $\mathbf{y} \in \{\text{Refute}, \text{Support}\}$, we follow Huang et al. (2024); Yu et al. (2024) and evaluate accuracy under three conditions. Let $\mathbf{e}_c$ and $\mathbf{e}_w$ denote correct and incorrect evidence, respectively, and let the model prediction be $\hat{\mathbf{y}} = f_\theta(\mathbf{c}, \mathbf{e})$. The indicator function $\mathbb{I}(\cdot)$ equals 1 if the argument is true and 0 otherwise. Then we define:

- Accuracy w/ Correct Evidence (CE): $\text{CE} = \frac{1}{|\mathcal{Q}|} \sum_{(\mathbf{c}, \mathbf{y}) \in \mathcal{Q}} \mathbb{I}(f_\theta(\mathbf{c}, \mathbf{e}_c) = \mathbf{y})$

- Accuracy w/ Wrong Evidence (WE): $\text{WE} = \frac{1}{|\mathcal{Q}|} \sum_{(\mathbf{c}, \mathbf{y}) \in \mathcal{Q}} \mathbb{I}(f_\theta(\mathbf{c}, \mathbf{e}_w) = \mathbf{y})$

- Overall Accuracy (OA): $\text{OA} = \frac{\text{CE}+\text{WE}}{2}$

**Main Results.** Our method demonstrates superior performance across diverse language models, achieving the best or competitive overall accuracy in five out of six model evaluations. The most notable advantage emerges on smaller-scale models like Mistral-7B, where our approach surpasses the

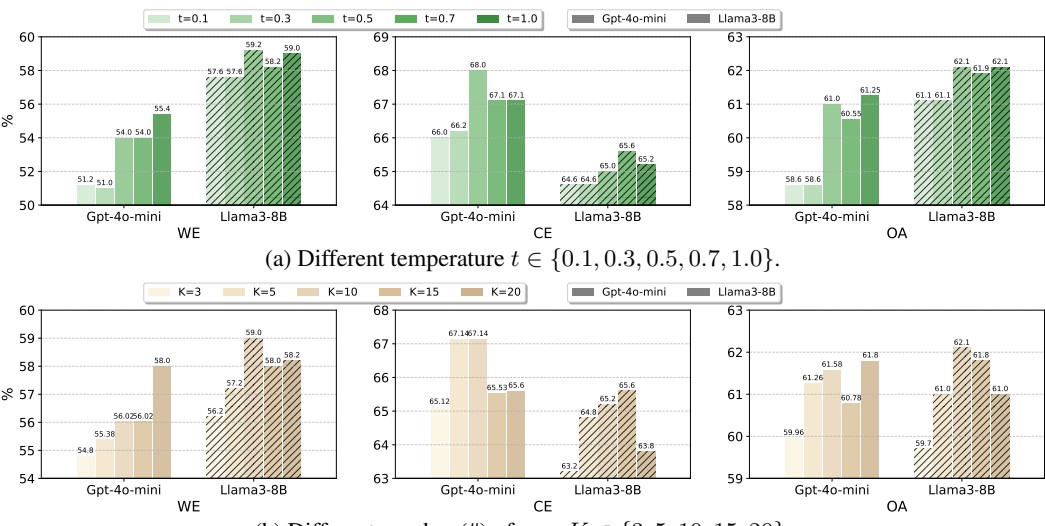

(a) Different temperature $t \in \{0.1, 0.3, 0.5, 0.7, 1.0\}$.

(b) Different number (#) of runs $K \in \{3, 5, 10, 15, 20\}$.

Figure 4: Comparison of our method across three metrics (WE, CE, and OA) in different numbers (#) of runs $K$ (a) and temperature (b) based on two LLMs: Gpt-4o-mini and Llama3-8B.

second-best method by 6.58 absolute points (57.08% vs 50.50%), highlighting its exceptional efficacy. Furthermore, while ContextEval remains a strong competitor, our method maintains balanced performance between CE and WE handling across most models, indicating robust arbitration capabilities rather than over-reliance on evidence quality or parametric probability. For instance, Gpt-4o-mini achieves optimal performance in both WE (55.38%) and CE (67.14%) scenarios, demonstrating consistent reliability. The results are examined on the FactConf dataset, with $K = 5$ runs and temperature 1.0.

**Analysis of efficiency and temperature.** We evaluate the performance of our method across different runs (K = 3, 5, 10, 15, 20) using Llama3-8B and Gpt-4o-mini as benchmark models. As shown in Figure 4 (b), further increases in the number of runs yield only marginal improvements, indicating that performance does not consistently improve or degrade with additional iterations. We select K=5 to balance accuracy and computational overhead. Our method also achieves competitive performance at K=3, which incurs the same overhead as the baseline method InternalEval (requiring only one additional inference per claim compared to other baselines). At $K = 3$, Llama3-8B achieves a WE of 56.20, surpassing the second-best value of 54.02, and a CE of 63.20, exceeding the second-best value of 61.17. These results demonstrate that our approach remains highly competitive even under constrained computational budgets, offering a practical alternative for users prioritizing efficiency. Regarding the impact of temperature, as illustrated in Figure 4 (a), experimental results on Llama3-8B and Gpt-4o-mini show that increasing the temperature from 0.1 to 0.5 leads to noticeable performance gains across WE, CE, and OA. However, no significant or consistent improvement is observed when the temperature rises from 0.5 to 1.0.

# 6 CONCLUSION

Our work establishes a comprehensive foundation for knowledge conflict evaluation in fact-checking through three key contributions. We introduce a novel fine-grained framework that yields critical insights into LLM behavior, particularly revealing that models are more susceptible to influence in the NOT-KNOW condition than in the KNOW condition, where they tend to persist with initial responses. By constructing specialized datasets from public benchmarks and *new* events for LLMs' outdated knowledge scenarios, we enable rigorous testing of models' evidence integration capabilities. Additionally, we develop a test-time method that effectively guides models in making faithful decisions between parametric knowledge and external knowledge, demonstrating superior performance over eight existing baselines.

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

## A    FURTHER DISCUSSION

Our evaluation with and without external evidence, based on the prompts in Table 16, and the entity substitution generation prompts are shown in Table 17.

### A.1    COUNTERFACTUAL CONFLICTING CONSTRUCTION.

We conduct controlled experiments to compare two prevalent methods for constructing conflicting evidence: the entity substitution approach(Longpre et al., 2021b) adopted in our work, and a more recent technique(Xie et al., 2023) that leverages LLMs to generate coherent counterfactual passages that contradict parametric knowledge. As illustrated in Figure 6, counterfactual evidence (CF in the figure) poses a significantly greater challenge for LLMs than entity-substituted evidence (ES in the figure), except for Llama3. This finding underscores the importance of pre-processing counterfactual conflicts before exposing them to the model. Our results indicate that adaptive counterfactual generation

Figure 5: Performance under conflicting external evidence (with a much more challenging constructed method), evaluated using influence rate (IR), persistence rate (PR), odds of influence (O), and sample count (#) in KNOW and NOT-KNOW states for initial CORRECT responses.

|  | **KNOW** | | | **NOT-KNOW** | | |
|---|---|---|---|---|---|---|
|  | IR | PR | O | IR | PR | O |
| Gpt-4o-mini | 91.47 | 8.53 | 10.72 | 96.92 | 3.08 | 31.47 |
| Qwen3-32B | 92.84 | 7.16 | 12.97 | 96.12 | 3.88 | 24.77 |
| Llama3-8b | 66.88 | 33.12 | 2.02 | 68.37 | 31.63 | 2.16 |
| Mistral-7b | 84.21 | 15.79 | 5.33 | 86.04 | 13.96 | 6.16 |
| Phi-4 | 99.87 | 0.13 | 768.23 | 99.93 | 0.70 | 142.76 |

introduces higher complexity, which may require additional computational strategies to mitigate its negative impact on model performance. Detailed experimental results are presented in Table 5. The prompts of counterfactual evidence (CF) are shown in the Table 18.

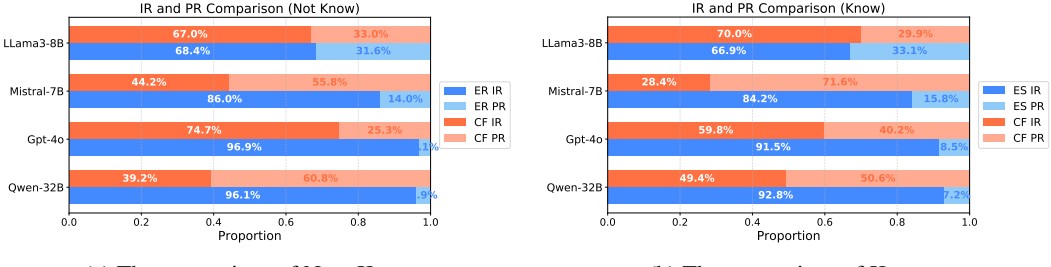

(a) The comparison of NOT-KNOW.

(b) The comparison of KNOW.

Figure 6: The comparison of influence rate (IR) and persistence rate (PR) with different levels of wrong conflict evidence in different LLMs. The entity substitution (ES) is highlighted in blue, and counterfactual evidence is highlighted in red.

### A.2    THE RATIO OF SUPPORT AND REFUTE

Table 6: The ratio (%) on support and refute ratios in Know and Not-Know categories, separated as initial CORRECT (Left) and WRONG (Right) responses.

|  | **Know** | | **Not-Know** | | **Know** | | **Not-Know** | |
|---|---|---|---|---|---|---|---|---|
|  | Support | Refute | Support | Refute | Support | Refute | Support | Refute |
| **Deepseek-v3** | 29.94 | 70.06 | 65.00 | 35.00 | 85.05 | 14.95 | 63.69 | 36.31 |
| **Gemini-2.5** | 40.88 | 59.12 | 60.79 | 39.21 | 66.92 | 33.08 | 62.01 | 37.99 |
| **Qwen3-32B** | 36.67 | 63.33 | 65.50 | 35.00 | 64.47 | 35.53 | 56.01 | 43.99 |
| **Gpt-4o-mini** | 32.88 | 67.12 | 70.89 | 29.11 | 79.67 | 20.33 | 66.38 | 33.62 |
| **Llama3-8B** | 31.47 | 68.53 | 45.35 | 54.65 | 81.18 | 18.82 | 61.82 | 38.18 |
| **Phi-4** | 26.39 | 73.61 | 42.05 | 57.95 | 71.30 | 28.70 | 61.39 | 38.61 |
| **Mistral-7B** | 14.46 | 85.54 | 43.26 | 56.74 | 80.60 | 19.40 | 62.12 | 37.88 |

In Table 6, we provide the support and refute ratio within Know and Not-Know. Our analysis yields several primary findings: 1) As shown in the Know columns under the "Correct" section, all

models demonstrate a much higher proportion of Refute cases when they answer correctly based on their internal knowledge. For instance, Deepseek-v3's correct Know answers are composed of 85.05% Refute statements, and most other models show a Refute ratio well above 60%. This strongly suggests that the reliable, pre-trained knowledge of LLMs is predominantly geared towards negation. In other words, models may be inherently more capable of confidently and correctly identifying a false statement than confirming a true one. 2)In the Not-Know columns for correct answers, Support cases consistently outnumber Refute cases (e.g., 70.89% support vs 29.11% refute of Gpt-4o-mini). This indicates that affirmative statements constitute the bulk of questions that models may be uncertain about and require external tools to verify.

## A.3 ANALYSIS OF DIFFERENT TEMPERATURES AND RUNS

We examine the sensitivity of model behavior to variations in decoding temperature, a key hyperparameter that controls the randomness of token sampling. As summarized in Table 7, we evaluate the Influence Rate (IR) and Persistence Rate (PR) across three temperature settings (T = 0.3, 0.5, 1.0) under both KNOW and NOT-KNOW conditions. The most salient finding is the remarkable stability of IR and PR metrics across temperature variations. For initially CORRECT responses, the IR remains consistently low (approximately 25%) while the PR remains high (approximately 75%), indicating strong resistance to conflicting evidence regardless of

Figure 7: Comparison across temperatures when facing conflict evidence in Gpt-4o-mini. The influence rate (IR %) and persistence rate (PR %) among different temperatures

|  | T=1.0 | | T=0.5 | | T=0.3 | |
|---|---|---|---|---|---|---|
|  | IR | PR | IR | PR | IR | PR |
| KNOW | | | | | | |
| CORRECT | 24.48 | 75.52 | 24.65 | 75.35 | 24.73 | 75.26 |
| WRONG | 61.97 | 38.03 | 61.53 | 38.47 | 59.77 | 40.23 |
| NOT-KNOW | | | | | | |
| CORRECT | 26.73 | 73.27 | 26.43 | 73.57 | 25.65 | 74.35 |
| WRONG | 62.57 | 37.43 | 62.48 | 37.52 | 60.82 | 39.18 |

temperature setting. Conversely, for initially WRONG responses, the IR remains consistently high (approximately 60-62%) with correspondingly low PR values (approximately 38-40%). Additionally, as shown in Figure 9, the impact of different temperatures on other evaluation metrics follows a similar trend, further confirming the robustness of model behavior to this hyperparameter.

We further assess the consistency of model performance across multiple independent runs. Figure 8 compares results from repeated evaluations using Llama3-8B, and Phi-4 with fixed temperature 0.3.

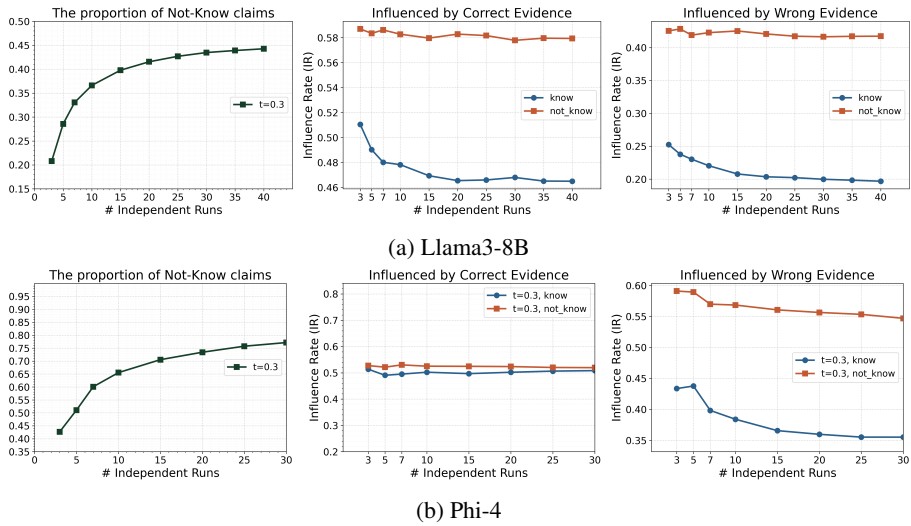

Figure 8: Comparison of independent runs under temperature (0.3) based on Llama3-8B and Phi-4.

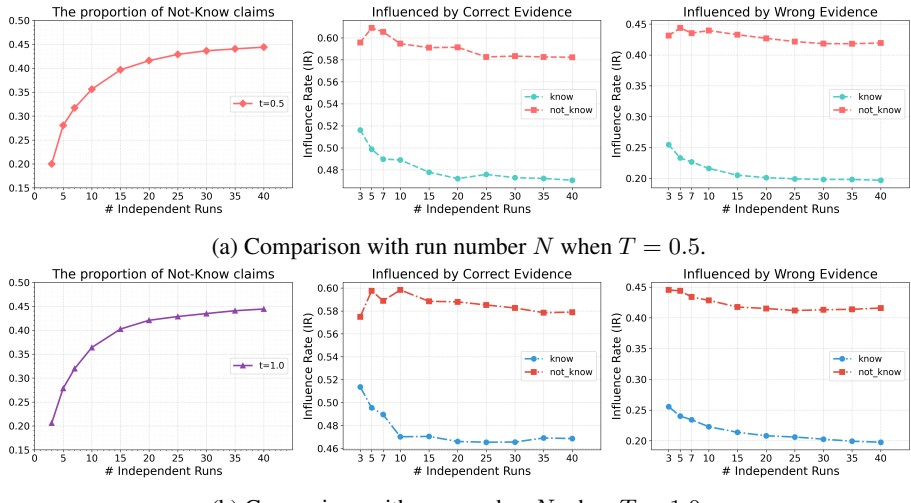

(a) Comparison with run number $N$ when $T = 0.5$.

(b) Comparison with run number $N$ when $T = 1.0$.

Figure 9: Comparison of run number $N$ under different temperature settings (Llama3-8B). Subfigures show results for: (a) $T = 0.5$, and (b) $T = 1.0$.

### A.4 THE VARIANCE OF EVALUATION RESULTS

For the elevation results of FactConf (Table 2), we present the variance and standard deviation statistics for the influence rate (IR) and persistence rate (PR) across several language models, under conditions where the initial response is either correct or incorrect, as well as for samples in the KNOW and NOT-KNOW groups. Shown in Tables 8 and 7. Overall, the results reveal clear differences in the stability of models' influence and persistence behaviors. Notably, LLama3:8b consistently demonstrates the lowest variance and standard deviation in both correct and incorrect response conditions, indicating a highly stable response pattern. In contrast, Qwen3-32b displays the highest variance and standard deviation values, suggesting that its behavior is substantially less consistent and more variable. Most models exhibit similar variance and standard deviation between influence rate and persistence rate, implying that the volatility in model output affects both metrics equally. It is also observed that, for many models (e.g., Gpt-4o-mini, Deepseek-v3, and emini-2.5-flash), the NOT-KNOW group tends to have higher variability when the initial response is incorrect, signifying greater unpredictability when the model lacks knowledge. In summary, LLama3 and Deepseek-v3 stand out as the most stable models among those evaluated, whereas Qwen3-32b is characterized by pronounced instability.

Table 7: Variance (Var) and Standard Deviation (Std) for influence rate (IR) and persistence rate (PR) KNOW and NOT-KNOW when initial response is CORRECT.

| Model | KNOW | | | | NOT-KNOW | | | |
|---|---|---|---|---|---|---|---|---|
| | IR Var | IR Std | PR Var | PR Std | IR Var | IR Std | PR Var | PR Std |
| Gpt-4o-mini | 0.0093 | 0.0967 | 0.0093 | 0.0967 | 0.0064 | 0.0799 | 0.0064 | 0.0799 |
| Deepseek-v3 | 0.0019 | 0.0435 | 0.0019 | 0.0435 | 0.0034 | 0.0585 | 0.0034 | 0.0585 |
| Qwen3-32B | 0.0451 | 0.2125 | 0.0451 | 0.2125 | 0.0450 | 0.2120 | 0.0450 | 0.2120 |
| Gemini-2.5-flash | 0.0129 | 0.1134 | 0.0129 | 0.1134 | 0.0064 | 0.0801 | 0.0064 | 0.0801 |
| Mistral-7B | 0.0053 | 0.0729 | 0.0053 | 0.0729 | 0.0035 | 0.0591 | 0.0035 | 0.0591 |
| Phi-4 | 0.0050 | 0.0704 | 0.0050 | 0.0704 | 0.0033 | 0.0575 | 0.0033 | 0.0575 |
| Llama3-8B | 0.0009 | 0.0293 | 0.0009 | 0.0293 | 0.0002 | 0.0135 | 0.0002 | 0.0135 |

### A.5 HALLUCINATION ANALYSIS

We analyze the results with an explanation based on GPT-4o-mini in outdated claims and results shown in Table 9. The LLM response explicitly states a knowledge cutoff date of October 2023 but proceeds to evaluate a claim involving a specific future date: May 18, 2025. This date corresponds

Table 8: Variance (Var) and Standard Deviation (Std) for influence rate (IR) and persistence rate (PR) of KNOW and NOT-KNOW when the initial response is WRONG.

| Model | KNOW | | | | NOT-KNOW | | | |
|---|---|---|---|---|---|---|---|---|
| | IR Var | IR Std | PR Var | PR Std | IR Var | IR Std | PR Var | PR Std |
| Gpt-4o-mini | 0.0034 | 0.0584 | 0.0034 | 0.0584 | 0.0087 | 0.0935 | 0.0087 | 0.0935 |
| Deepseek-v3 | 0.0006 | 0.0244 | 0.0006 | 0.0244 | 0.0060 | 0.0774 | 0.0060 | 0.0774 |
| Qwen3-32B | 0.0521 | 0.2283 | 0.0521 | 0.2283 | 0.0569 | 0.2386 | 0.0569 | 0.2386 |
| Gemini-2.5-flash | 0.0024 | 0.0495 | 0.0024 | 0.0495 | 0.0049 | 0.0699 | 0.0049 | 0.0699 |
| Llama3-8B | 0.0009 | 0.0293 | 0.0009 | 0.0293 | 0.0002 | 0.0135 | 0.0002 | 0.0135 |
| Mistral-7B | 0.0009 | 0.0301 | 0.0009 | 0.0301 | 0.0009 | 0.0293 | 0.0009 | 0.0293 |
| Phi-4 | 0.0046 | 0.0678 | 0.0046 | 0.0678 | 0.0038 | 0.0616 | 0.0038 | 0.0616 |

to the purported final Everton F.C. match at Goodison Park. While the model correctly references Everton's publicly known stadium relocation plans and the typical timing of the Premier League season's conclusion, its concrete acceptance of May 18, 2025 as a plausible final match date exceeds verified knowledge boundaries. Specifically, since official fixture lists and exact match dates for the 2024–25 Premier League season would likely only be confirmed and publicly released in mid-2025, the model's certainty about the precise date is unsupported by direct evidence available before the cutoff. The response frames the date as "plausible" but ultimately judges the claim as "should be supported," which risks conflating reasonable inference with factual confirmation.

This constitutes a form of *hallucination*, where the model goes beyond its verified knowledge to make a detailed prediction without authoritative backing. Although not fabricated outright, it represents an overconfident extrapolation that could mislead users into treating the claim as a confirmed fact rather than an informed speculation.

Table 9: Data instance with response based on Gpt-4o-mini.

| Claim | LLM Result |
|---|---|
| On May 18 2025: 2024–25 Premier League Everton F.C. play their final game at Goodison Park in Walton, Liverpool, England, the club's home ground for the last 133 years. They will transition to Everton Stadium in Vauxhall starting the 2025–26 season. (BBC Sport) | The claim should be supported based on the information available up to October 2023. Explanation: Everton FC has been planning to move to a new stadium known as Everton Stadium which is being constructed in the Vauxhall area of Liverpool. The club has announced that they intend to transition to this new stadium, which is expected to be completed in time for the 2025–26 season. As of my last update, the club had been preparing for this significant move, which would mark the end of their long history at Goodison Park, their home since 1892. The specific date mentioned in the claim (May 18, 2025) aligns with the conclusion of the 2024–25 Premier League season, making it plausible that this could be the date of their final game at Goodison Park. Therefore, the claim is consistent with the club's announcements and plans regarding their stadium transition. |

# B  RELATED WORKS

Following the taxonomy established in the survey (Xu et al., 2024), knowledge conflicts can be broadly categorized into three types: (1) Context-Memory Conflict, where contextual knowledge (e.g., user prompts, dialogue history, or retrieved documents) conflicts with the model's internal parametric knowledge; (2) Inter-Context Conflict, which arises from inconsistencies among different pieces of retrieved evidence; and (3) Intra-Memory Conflict, stemming from inherent inconsistencies present within the pretraining data itself.

- Intra-memory and Inter-context Conflicts. ConflictBank (Su et al., 2024) is a benchmark primarily focused on conflicts arising from intra-memory and inter-context issues. However, our paper targets the context-memory conflict specifically. While ConflictBank is organized around the causes of conflicts, such as misinformation, our evaluation framework is structured around the internal knowledge state of the model (Know vs. Not-Know). This approach allows us to directly assess whether large language models (LLMs) possess a clear self-awareness of their knowledge boundaries, which is a crucial factor for reliable fact-checking that previous benchmarks do not address. Pham et al. (2024)introduced the WhoQA benchmark to investigate how LLMs behave when retrieved documents contain conflicting information about entities sharing a name. Their work advocates for a model behavior of transparently informing users about conflicts rather than

making an autonomous, potentially biased decision. The WhoQA benchmark is designed to systematically induce such conflicts, revealing that even simple questions can significantly degrade LLM performance in RAG settings when conflicts are present. In parallel, Gao et al. (2025) address the faithfulness issue in Retrieval-Augmented Generation (RAG) under knowledge conflicts. They propose CLEAR, a method that probes internal hidden states to localize conflicts and employs conflict-aware fine-tuning to guide more accurate evidence integration. Wang et al. (2025) analyze knowledge conflicts from an information-theoretic perspective and introduces Swin-VIB, a novel framework that enhances LLMs' robustness by adapting retrieved information.

Our work specifically falls under the first category—Context-Memory Conflict. We focus on the scenario where external contexts conflict with the model's parametric knowledge, and investigate how the model's awareness of its own knowledge boundaries influences its response to such conflicts.

## C    ADDITIONAL DETAILS AND RESULTS OF OUR METHOD

### C.1    THE IMPLEMENTATION DETAILS

Our method aims to assess the internal consistency of a model's answers by leveraging the answer probability associated with each response and the algorithm shown in Algorithm 1.

---

**Algorithm 1** Evidence Stability via Jensen-Shannon Divergence

---

**Require:** Claim $\mathbf{c}$, Evidence $\mathbf{e}$, an LLM $f_\theta$, iteration times $K$, Threshold $\tau$
**Ensure:** Decision: Provide the answer $\hat{\mathbf{y}}$ by using or not using external evidence $\mathbf{e}$

1: Initialize arrays $\{p_{\text{ext}}^{(i)}\}_{i=1}^{K}$, $\{p_{\text{par}}^{(i)}\}_{i=1}^{K}$
2: **for** $i = 1$ to $K$ **do**
3:     $p_{\text{ext}}^{(i)} \leftarrow f_\theta(\mathbf{c}, \mathbf{e})$     ▷ Sample with evidence
4:     $p_{\text{par}}^{(i)} \leftarrow f_\theta(\mathbf{c})$     ▷ Sample without evidence
5: **end for**
6: Compute mean probabilities: $\bar{p}_{\text{ext}} \leftarrow \frac{1}{K} \sum_{i=1}^{K} p_{\text{ext}}^{(i)}, \quad \bar{p}_{\text{par}} \leftarrow \frac{1}{K} \sum_{i=1}^{K} p_{\text{par}}^{(i)}$
7: Compute evidence stability: $\mathbf{JSD}_{\text{ext}} \leftarrow H(\bar{p}_{\text{ext}}) - \frac{1}{K} \sum_{i=1}^{K} H(p_{\text{ext}}^{(i)})$     ▷ Using the binary entropy function
8: Compute parametric stability: $\mathbf{JSD}_{\text{par}} \leftarrow H(\bar{p}_{\text{par}}) - \frac{1}{K} \sum_{i=1}^{K} H(p_{\text{par}}^{(i)})$     ▷ Using the binary entropy function
9: $\Delta = \mathbf{JSD}_{\text{par}} - \mathbf{JSD}_{\text{ext}}$
10: **if** $\Delta \leq \tau$ **then**
11:     Do not use evidence, **return** $\hat{\mathbf{y}} = f_\theta(\mathbf{c})$     ▷ External evidence introduces variability
12: **else**
13:     Use evidence, **return** $\hat{\mathbf{y}} = f_\theta(\mathbf{c}, \mathbf{e})$
14: **end if**

---

As shown in Table 5. Our method achieves competitive performance across six diverse language models, demonstrating remarkable generalization capability. It attains the best overall accuracy on Gpt-4o-mini, Mistral-7B, Qwen3-32B, and Llama3-8B, while securing second place on Phi-4. The most striking advantage emerges on smaller-scale models: on Mistral-7B, our approach outperforms the second-best method by 6.58 absolute percentage points, revealing its particular efficacy in resource-constrained environments. This scalability advantage is crucial for practical deployments where computational efficiency is paramount. Furthermore, our method maintains balanced performance between correct evidence (CE) and wrong evidence (WE) handling across most models, indicating robust reasoning capabilities rather than over-reliance on evidence quality. The sole exception occurs with Gemini-2.5, where ExplicitSCR performs best, suggesting model-specific optimization opportunities. We discuss the influence of different iteration times as follows.

We run train-test experiments three times to set hyperparameters $\tau$. Each time renders an optimal $\tau$ that ranges in $[-0.08, -0.06]$, in which $\tau$ is determined using a small set of held-out validation data.

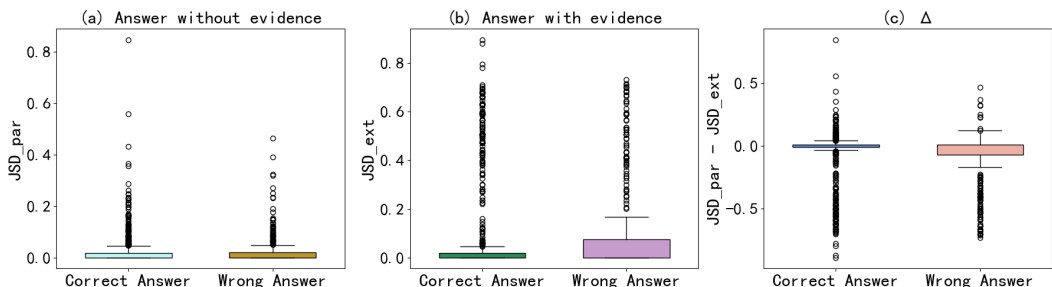

Figure 10: The comparison of our defined JSD with and without evidence.

As shown in Figure 10, we compare the Jensen–Shannon Divergence (JSD) $\mathbf{JSD} = H(\bar{p}) - \mathbb{E}[H(p)]$ with correct evidence under two settings: answering with the claim only (Figure (a)) and answering with both the claim and external evidence (Figure (b)), using GPT-4o-mini under correct and incorrect response conditions. In the claim-only scenario (Figure (a)), the JSD values are similar regardless of whether the model's answer is correct or incorrect. In contrast, when external evidence is provided (Figure (b)), the JSD is noticeably lower when the model's answer is correct than when it is incorrect. This pattern suggests that GPT-4o-mini may possess a certain capability to discern evidence quality: while external evidence may still influence the model, it also introduces greater internal uncertainty, as reflected by the elevated JSD in incorrect responses.

## C.2 THE VOTED-BASED JSD

While Our method aims to assess the internal consistency of a model's answers by leveraging the answer probability associated with each response, not all models provide direct access to reliable answer probabilities. To address this limitation, we introduce a simple variant of our approach that estimates stability using the results of multiple consistency checks for models that can not access the probability directly.

Specifically, we approximate the external stability measure $\mathbf{JSD} = H(\bar{p}) - \mathbb{E}[H(p)]$ using vote entropy—a frequency-based entropy computed from hard labels. Formally, given a set of $K$ labels (in $K$ runs), let $k_{\text{true}} \in [0, K]$ denote the number of labels that belong to any of the true categories. Then, the empirical probability is estimated as $\hat{p} = k_{\text{true}}/K$, and the vote entropy is computed as the binary entropy of $\hat{p}$: $H_{\text{binary}}(\hat{p}) = -\hat{p}\log_2(\hat{p}) - (1 - \hat{p})\log_2(1 - \hat{p})$ This variant allows us to robustly evaluate stability even when explicit probability outputs are unavailable.

## C.3 CASE STUDY

To empirically validate the effectiveness of our stability-based evidence selection framework, we present a comprehensive case study analyzing three distinct factual verification scenarios. These cases demonstrate how our method dynamically arbitrates between parametric knowledge and external evidence based on output stability metrics. As shown in Table 10.

The first case involves a claim regarding high school arrest statistics and graduation rates, where the provided external evidence contains internal contradictions (marked with $\times$). Our method computes a negative $\Delta$ value ($-0.2229$), correctly identifying that the evidence introduces substantial variability ($\mathbf{JSD}_{\text{par}} = 0.2239$) compared to the highly stable parametric responses ($\mathbf{JSD}_{\text{ext}} = 0.0011$). Consequently, the system rejects the conflicting evidence and maintains the original "support" classification based solely on parametric knowledge.

In the second case examining trade agreement impacts on employment, the external evidence provides consistent factual support (marked with $\checkmark$). Here, the parametric responses show significant instability ($\mathbf{JSD}_{\text{ext}} = 0.1438$) with mixed "refute"/"support" predictions, while evidence-augmented queries yield stable consensus ($\mathbf{JSD}_{\text{par}} = 0.0165$). The positive $\Delta$ ($0.1273$) correctly triggers evidence utilization, shifting the majority prediction from "refute" to "support."

The third case, concerning national debt growth under different presidential administrations, further illustrates the method's precision. Despite initial parametric uncertainty (mixed responses with

Table 10: Here are examples of our method for selecting answers, with and without supporting evidence, demonstrated across 10 independent runs. Our approach can successfully identify the correct initial answer even when the external evidence is inaccurate (indicated by ×). Additionally, it selects the answer backed by evidence when that evidence is correct (indicated by ✓)

.

| | |
|---|---|
| Claim (**c**) | High school students arrested on campus are twice as likely not to graduate and four times less likely to graduate if they've appeared in court. |
| External Evidence **e** | high school dropouts are three and one-half times more likely than high school graduates to be arrested, and more than eight times as likely to be incarcerated. 12 pages, seventy-five middle school students who were under court have been involved in the judicial system, making sure they "complete college". High school dropouts are three and one-half times more likely than high school graduates to be arrested, and more than eight times as likely to be incarcerated. × |
| Initial Answer $f_\theta(\hat{\mathbf{y}} \mid \mathbf{c}_i)$ | "support", "support", "support", "support", "support", "support", "support", "support", "support", "support" |
| Initial Answer Probabilities | 0.9965, 0.9950, 0.9971, 0.9858, 0.9950, 0.9965, 0.9965, 0.9965, 0.9971, 0.9958 |
| Answer with evidence $f_\theta(\hat{\mathbf{y}} \mid \mathbf{c}_i, \mathbf{e}_i)$ | "refute", "refute", "refute", "refute", "refute", "refute", "refute", "refute", "refute", "refute" |
| Evidence Probabilities | 0.9399, 0.9241, 0.4115, 0.4620, 0.9399, 0.9399, 0.8808, 0.4620, 0.9399, 0.4046 |
| **JSD**$_{\text{ext}}$ (claim only) | 0.0011 |
| **JSD**$_{\text{par}}$ (claim with conflicting evidence) | 0.2239 |
| $\Delta = \textbf{JSD}_{\text{ext}} - \textbf{JSD}_{\text{par}}$ | ≈ −0.2229 (Choose answer with claim only) |
| Claim (**c**) | Trade agreements like NAFTA and permanent normal trade relations with China, which forced American workers to compete against people making pennies an hour, has resulted in the loss of 160,000 jobs here in Michigan." |
| External Evidence **e** | trade agreements like NAFTA and permanent normal trade relations with China, which forced American workers to compete against people making pennies an hour, have resulted in the loss of 160,000 jobs here in Michigan. The steel industry setbacks account for just a fraction of the job losses in Michigan's manufacturing sector, which now employs 55,100 fewer workers than it did when Trump took office in January 2017, U.S. Labor Department data shows. The state's automotive industry accounted for 35% of the manufacturing job losses, according to the st. Louis fed. Trade agreements like NAFTA and permanent normal trade relations with China, which forced American workers to compete against people making pennies an hour, have resulted in the loss of 160,000 jobs here in Michigan. ✓ |
| Initial Answer (10 times) | "refute", "refute", "refute", "refute", "support", "refute", "support", "support", "refute", "support" |
| Initial Answer Probabilities $f_\theta(\hat{\mathbf{y}} \mid \mathbf{c})$ | 0.9193, 0.6770, 0.6183, 0.6171, 0.3707, 0.6770, 0.8121, 0.3707, 0.6199, 0.8121 |
| Answer with Evidence $f_\theta(\hat{\mathbf{y}} \mid \mathbf{c}_i, \mathbf{e})$ | "support", "refute", "support", "support", "support", "support", "support", "support", "support", "support" |
| Evidence Probabilities | 0.8175, 0.1480, 0.8519, 0.9526, 0.8175, 0.9526, 0.8519, 0.9046, 0.8519, 0.9046 |
| **JSD**$_{\text{ext}}$ (claim only) | 0.1438 |
| **JSD**$_{\text{par}}$ (claim with conflicting evidence) | 0.0165 |
| $\Delta = \textbf{JSD}_{\text{ext}} - \textbf{JSD}_{\text{par}}$ | ≈ 0.1273 (Choose answer with evidence) |
| Claim (**c**) | Says under President Barack Obama, the debt increased by 23 percent, which was less than under any president going back to Ronald Reagan. |
| External Evidence (**e**) | 27 Feb 2023 under President Barack Obama, the national debt grew from $10.63 to $19.96 trillion, a 87.8% increase. Under President Donald Trump, the ... Oct 7, 2022, Bush nearly matched the amount of debt accumulated under Reagan, but ... Bush, democratic president barack obama added another $8.34 trillion ... May 15, 2023, Ronald Reagan and George W. Bush. President Ronald Reagan increased the U.S. debt by around 1.86 trillion U.S. dollars, or 186.36 percent. this ... ✓ |
| Initial Answer (10 times) | "refute", "support", "refute", "support", "refute", "support", "refute", "support", "refute", "support" |
| Initial Answer Probabilities $f_\theta(\hat{\mathbf{y}} \mid \mathbf{c}_i)$ | 0.3760, 0.8485, 0.3760, 0.6764, 0.3760, 0.6183, 0.3760, 0.8485, 0.4360, 0.7230 |
| Answer with Evidence $f_\theta(\textbf{haty} \mid \mathbf{c}_i, \mathbf{e}_i)$ | "refute", "refute", "refute", "refute", "refute", "refute", "refute", "refute", "refute", "refute" |
| Evidence Probabilities | 0.9999973, 0.9999973, 0.9999973, 0.9999971, 0.9999973, 0.9999973, 0.9999981, 0.9999973, 0.9999971, 0.9999971 |
| **JSD**$_{\text{ext}}$ (claim only) | 0.0316 |
| **JSD**$_{\text{par}}$ (claim with conflicting evidence) | 2.5305e-08 |
| $\Delta = \textbf{JSD}_{\text{ext}} - \textbf{JSD}_{\text{par}}$ | ≈ 0.0316 (Choose answer with evidence) |

**JSD**$_{\text{ext}} = 0.0316$), the external evidence produces near-perfect stability (**JSD**$_{\text{par}} \approx 2.53 \times 10^{-8}$). The positive $\Delta$ (0.0316) leads to evidence adoption, resulting in unanimous "refute" decisions that correctly contradict the original claim.

These cases collectively demonstrate three critical capabilities: (1) the ability to detect and reject unreliable evidence that introduces instability; (2) the capacity to identify and utilize high-quality evidence that enhances prediction consistency; and (3) the robustness to make appropriate evidence selection decisions across diverse factual domains without requiring explicit credibility assessments

of the evidence content. The stability metric $\Delta$ serves as an effective proxy for evidence quality, enabling reliable arbitration between internal knowledge and external information sources.

## D   THE DETAILS OF DATASETS

To ensure the quality of automatically constructed conflicting evidence, we established a multi-stage quality control process to verify its plausibility and reliability. The generated evidence underwent systematic filtering and validation under multiple constraints: during generation, the LLM was instructed to ensure that substituted entities shared the same type as the original and that at least ten key entities could be extracted from the context to guarantee semantic richness. After automatic construction, we conducted manual sampling reviews by three human annotators. This rigorous workflow ensures the high quality and challenging nature of the final dataset.

### D.1   NEWFACTCONF DATASET

Table 11: Example of our collected events with evidence, which is new to Phi-4, Deepseek-v3, etc.

| Claim | Evidence | Time | Outdated for LLMs |
|---|---|---|---|
| July 31,2025. The military junta of Myanmar formally ends the country's four-year-long state of emergency and declares a December 2025 election for the country's new head of government and legislative members. | ... Myanmar's military government plans to hold a general election for elected seats in the Amyotha Hluttaw and the Pyithu Hluttaw of the Assembly of the Union, currently dissolved, on a date in December 2025 to be determined. The planned election would be the first after the 2021 military coup d'état ... | July 31, 2025 | Gpt-4o-mini, LLama3-8B Phi-4 Deepseek-v3, Gemini-2.5-flash Mistral-7B |

Our constructed dataset **NewFactConf**, which contains 3,995 true claims, is used to analyze LLM performance on unseen emerging events; the examples are shown in Table 11 and Table 15. Concretely, we collected *new* events as claims by crawling the Wikipedia current events portal in different months in 2024 and 2025 (e.g., `https://en.wikipedia.org/wiki/Portal:Current_events/June_2025`). If one LLM, such as Llama3-8B, which is published in April 2024, then the parametric knowledge is deemed to be unseen for the events that occurred in June 2025. So we annotated the events which crawled from the web page June 2025 is *new* to Llama3-8B. Then, as shown in Table 14, we systematically created conflicting evidence for

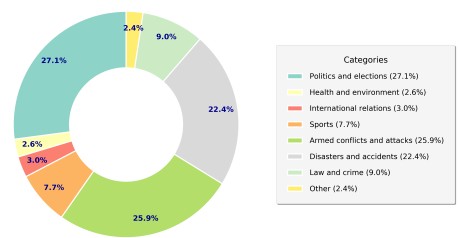

Figure 11: The categories of our NewFact-Conf.

every claim using the entity-substitution protocol described in previous experiments. The category distributions are displayed in Figure 11, and all the categories are sourced from Wikipedia.

Table 12: Comparison of Relative Odds (O′) across models on NewFactConf, showing O′ for initial CORRECT and WRONG responses.

| Relative Odds (O') | Gpt-4o-mini | Deepseek-v3 | Qwen3-32B | Gemini-2.5 | Llama3-8B | Mistral-7B | Phi-4 |
|---|---|---|---|---|---|---|---|
| CORRECT | 1156.7% | 825.1% | 146.3% | - | 350% | 502.3% | - |
| WRONG | 515.6% | 146.6% | -61.5% | -19.9% | -13.21% | 768.8% | 170.3% |

### D.2   FACTCONF DATASET

**The details of three based datasets.**   **QuanTemp** (Anand et al., 2024) is a diverse, multi-domain dataset exclusively focused on numerical claims. It encompasses comparative, statistical, interval, and temporal aspects, and includes detailed metadata along with a supporting evidence collection.

**Snopes**: We use the dataset from Popat et al. (2017), which consists of rumors from the general fact-checking website Snopes (www.snopes.com). Each instance includes an editor-assigned credibility label (true or false), alongside associated reporting articles and their sources.

**PolitiFact**: Popat et al. (2018) extracted a dataset from PolitiFact (www.politifact.com), which focuses on claims made by U.S. political figures. Our dataset includes all articles published prior to December 2017, each containing the claim, the speaker, and an official credibility rating.

We provide more examples of conflicting external evidence of FactConf in Table 13.

Table 13: Examples of conflicting external evidence of FactConf.

| Claim (c) | Label (y) | Evidence (e) | Constructed Conflicting Evidence (ē) |
|---|---|---|---|
| No member American public ever killed commercial nuclear power record unmatched fuels | True | ... nuclear energy is as safe or safer than any other form of energy available no member of the public has ever been injured or killed in the entire history of commercial nuclear power in the US. In fact recent studies have shown that it is safer to work in a nuclear power plant than an office ... | ... solar energy is as safe or safer than any other form of energy available no member of the public has ever been injured or killed in the entire history of renewable energy sources in the US. In fact recent studies have shown that it is safer to work in a solar power station than an office ... |
| slim jims recently cost 99 cents cost 269 | False | ...may thursday may 12th obama s health care law represents a government takeover of health care june friday july 15th 2011 a clause hidden in the obamacare bill which is now law gives obama the right to form a private army july sunday july 10th 2011 slim jims just recently cost 99 cents but now cost 269 august thursday august 11th 2011 we re inches away from no longer having a free economy september monday september 12th 2011 the first round of stimulus it created zero jobs october tuesday october 11th 2011 the irs is already planning on new employees... | ...may thursday may 12th obama s health care law represents a government takeover of health care june friday july 15th 2011 a clause hidden in the obamacare bill which is now law gives obama the right to form a private army july sunday july 10th 2011 beef jerky just recently cost Dollar Tree but now cost 370 august thursday august 11th 2011 we re inches away from no longer having a free economy september monday september 12th 2011 the first round of stimulus it created zero jobs october tuesday october 11th 2011 the irs is already planning on new employees... |

To ensure the quality of datasets, we implemented a multi-stage quality control process to ensure the plausibility and reliability of the generated conflicting evidence. This was not merely accepted but was systematically filtered and validated: During generation, we enforced several constraints, such as ensuring the substituted entity was of the same type as the original and that at least 5 key entities could be extracted from the context to ensure sufficient complexity. After the automated generation, we conducted a manual sampling review of the data. This human-in-the-loop step was crucial for verifying that the generated conflicts were coherent, plausible, and did not contain obvious artifacts. This process ensures the quality and challenge of our dataset.

## E  BASELINE METHODS

We compare several methods for knowledge conflicts attribution. Certain baselines, including ContextConf, TPC, and InternalConf, require access to the LLM's token-level probability estimates. Consequently, these methods are evaluated only on GPT-4o-mini, Phi-4, and Mistral-7B, from which such probability data can be obtained. The comprehensive results are presented in Table 5.

- **Implicit Self-Guided Confidence Reasoning (ImplicitSCR):** The model is explicitly instructed that the provided context may contain inaccuracies and must exercise independent judgment to evaluate its reliability before formulating a response. To prioritize the activation of internal knowledge, the context is deliberately positioned after the question. Some works (Huang et al., 2024) demonstrate that this sequential structure increases the model's reliance on prior knowledge, thereby reducing its vulnerability to misleading contextual information. The model subsequently engages in implicit confidence reasoning during its decision process and returns only the final answer. Prompts shown in Table 19.

- **Explicit Self-Guided Confidence Reasoning (ExplicitSCR):** In this approach, the model initially generates distinct responses derived from its internal knowledge and the provided context, using two separate prompts. It subsequently engages in verbalized confidence estimation through a chain-of-thought reasoning process. This begins with an evaluation of the confidence in its internally generated answer, including reflection on the factual basis underlying that response. The

Table 14: Example of conflicting external evidence of NewFactConf.

| Claim (c) | Label (y) | Evidence (e) | Constructed Conflicting Evidence (ē) |
|---|---|---|---|
| On April 6 2025: Russian invasion of Ukraine. 2025 Sumy Oblast incursion Russian troops reportedly capture the village of Basivka in Sumy Oblast, Ukraine. (Reuters) Kyiv strikes A Russian airstrike in Darnytskyi District, Kyiv, Ukraine, kills one person and injures three others. (CTV News) Kryvyi Rih strikes The death toll from Friday's missile strike on Kryvyi Rih, Ukraine, rises to 20 deaths, including several children, and 75 injuries. (CTV News) | True | ... In an effort to counter Ukraine's 2024 offensive in Kursk, in early 2025 Russian forces launched a cross-border offensive from Russia into Ukraine's adjacent Sumy Oblast. ... | ... In an effort to counter Poland's "2026 offensive" in Voronezh, in early 2026 Russian forces launched a "counter-terrorism operation" from Germany into Poland's adjacent Cherkasy Oblast. ... |
| On April 6 2025: Red Sea crisis March–April 2025 United States attacks in Yemen. Between four and 70 people are killed and at least 16 others are injured in overnight U.S. airstrikes targeting Houthi forces in Saada, Yemen. (CTV News) | True | ... In March 2025, the United States launched a large campaign of air and naval strikes against Houthi targets in Yemen. Code-named Operation Rough Rider, it has been the largest U.S. military operation in the Middle East of President Donald Trump's second term ... | ... In March 2026, the United States launched a large campaign of air and naval strikes against Houthi targets in Oman. Code-named Operation Iron Stallion, it has been the largest U.S. military operation in the Middle East of President Joe Biden's second term ... |

model then assesses the reliability of the external context by comparing it against the facts supporting its internal knowledge. The final answer is selected through a deliberative reasoning process that integrates both internal certainty and contextual credibility. Prompts shown in Table 19.

- **Internal Evaluation (InternalEval):** The LLM assesses the correctness of its internal answer through self-evaluation using the prompt provided in Table 20. If the self-evaluation confirms the answer is correct, the internal answer is retained; otherwise, the context-based answer is selected.

- **Context Evaluation (ContextEval):** The model evaluates the relevance and accuracy of the provided context in relation to the question (see Table 20 for the prompt specification). If the context is judged to be correct and reliable, the context-based answer is adopted; otherwise, the model defaults to its internal knowledge-based response.

- **Internal Confidence Thresholding (InternalConf):** The model selects its internal knowledge-based answer if its confidence exceeds a predefined threshold; otherwise, it defaults to the context-based answer. The threshold can be set to a fixed value (e.g., 0.5) or calibrated on a held-out dataset when available. Confidence estimates may be derived from sequence probabilities. Following ActiveRAG (Jiang et al., 2023a), Huang et al. (2024) simplifies to this internal confidence mechanism (InternalConf) to employ answer-level probability, yielding modest but consistent empirical improvements.

- **Context Confidence Thresholding (ContextConf):** The model evaluates the reliability of the external context. If confidence in the context exceeds a predefined threshold, the context-based answer is selected; otherwise, the model defaults to its internal knowledge-based response. The threshold can be determined using the same methodology as in Internal Confidence estimation. Confidence scoring may be derived from the sequence probability of the answer given the context.

- **(Calibrated) Token Probability Correction (TPC):** Following (Wu et al., 2024), compare the confidence scores—specifically, the mean token probabilities—of the model's internal answer and the context-based answer, selecting the one with the higher value as the final answer. This approach is termed token probability correction.

- **Truth-Aware Context Selection (TACS-LR)** Following (Yu et al., 2024), who provide Truth-Aware Context Selection (Yu et al., 2024), a context evaluation method, which employs a classifier to filter out incorrect content at a granular sentence or token level. Unlike rule-based approaches that simply choose between internal and context-based answers, this method reintegrates the filtered context into the LLM to regenerate the final answer. Huang et al. (2024) adopts this to an alternative strategy: the model explicitly removes untruthful sentences (TACS-LR; see Table 21 for the prompt), and the refined context is used to produce the final answer.

Table 15: We present examples of crawled events with evidence from NewFactConf. This information is new for some LLMs, indicating that their parametric knowledge is outdated.

| Claim | Evidence | Time | Outdated for LLMs |
|---|---|---|---|
| July 31, 2025. The military junta of Myanmar formally ends the country's four-year-long state of emergency and declares a December 2025 election for the country's new head of government and legislative members. | ...Myanmar's military government plans to hold a general election for elected seats in the Amyotha Hluttaw and the Pyithu Hluttaw of the Assembly of the Union, currently dissolved, on a date in December 2025 to be determined. The planned election would be the first after the 2021 military coup d'état... | July 31, 2025 | Deepseek-v3, Gpt-4o-mini, Llama3-8B Phi-4 Gemini-2.5-flash Mistral-7B |
| 2025 FIFA Club World Cup. In association football, English club Chelsea F.C. win the FIFA Club World Cup for the second time, defeating French club Paris Saint-Germain 3–0 in the final. Chelsea winger Cole Palmer is awarded the tournament's Golden Ball. (FIFA) | The 2025 FIFA Club World Cup final was the final match of the 2025 FIFA Club World Cup, the 21st edition of the premier competition for men's club soccer teams organized by FIFA. The match was played at MetLife Stadium at the Meadowlands Sports Complex in East Rutherford, New Jersey, near New York City, on July 13, 2025. It was contested between English club Chelsea and French club Paris Saint-Germain. This final was the first since 2000 to be contested by two teams from the same confederation—the only previous Club World Cup to feature multiple teams from the same association or country—and the first ever to feature two European teams. | July 13, 2025 | Deepseek-v3, Gpt-4o-mini, Llama3-8B Phi-4 Gemini-2.5-flash Mistral-7B |
| 2024 Maltese presidential election. Myriam Spiteri Debono is sworn in as President of Malta, succeeding George Vella and becoming the third woman to hold the office. (Times of Malta) | The 2024 Maltese presidential election took place on 27 March 2024. Members of the Parliament of Malta voted in an indirect election to elect the next President of Malta with former parliament speaker Myriam Spiteri Debono being the only nominee | April 4, 2024 | Mistral-7B |
| Arab–Israeli conflict.Gaza war.Gaza war hostage crisis. Hamas releases a propaganda video showing Israeli American hostage Edan Alexander. (CBS News) .An Israeli airstrike on a car in the Gaza Strip kills five people, including employees of World Central Kitchen. (AP) .Israel–Hezbollah conflict. Two people are killed and six others are injured in three airstrikes by Israel in southern Lebanon that occurred despite the ceasefire with Hezbollah. (Al Jazeera) | On 27 November 2024, a ceasefire agreement was signed by Israel, Lebanon, and five mediating countries, including the United States.[1] Hezbollah attacked Israel on 8 October 2023, leading to a year of cross-border fighting, and on 1 October 2024, Israel invaded Lebanon. The agreement mandates a 60-day halt to hostilities, during which Israel must withdraw its forces from Southern Lebanon,[2][3][4] and Hezbollah must withdraw its forces to north of the Litani River.[5] A five-country monitoring panel, led by the United States, would oversee the implementation, with 5,000 Lebanese troops deployed to ensure compliance.[6][3] The agreement does not preclude either Israel or Lebanon from acting in self-defense, but Israeli and Lebanese officials disagreed with what that entails.[7] Since the ceasefire went into effect, Lebanese sources claim Israeli attacks on Lebanon killed at least 83 civilians, while Israel said dozens of Hezbollah fighters were killed in the midst of ceasefire violations.[8][9] On 26 January 2025, the U.S. extended the agreement until 18 February.[10] Once this deadline lapsed, Israel withdrew from populated areas in southern Lebanon but declared that it would temporarily remain in five strategicL̈ebanese positions along the border | November 30, 2024 | Gpt-4o-mini Llama3-8B Mistral-7B |

## F  EVALUATION PROMPTS

We provide the prompts of basic evaluation and entity substitution in Table 16 and 17.

## G  LLM USAGE CLAIM

In this paper, LLMs are utilized exclusively for the purpose of aiding and polishing writing. Their application is strictly confined to improving linguistic clarity, coherence, grammar, and style within textual content. No additional functionalities are incorporated.

Table 16: The basic prompts of evaluation.

| Parametric or External Knowledge |
|---|
| **Parametric Knowledge Assessment (Without Evidence)**
Based solely on your internal knowledge, determine whether the claim should be supported or refuted. Provide only "support" or "refute" as the final answer without any explanation.

**Task:**
Claim: {claim}
Answer: |
| **External Knowledge Integration (With Evidence)**
Based on the provided information and your internal knowledge, determine whether the claim should be supported or refuted. Provide only "support" or "refute" as the final answer without any explanation.

**Task:**
Claim: {claim}
Information: {evidence}
Answer: |

Table 17: The prompts of entity substitution

| Entity Substitution |
|---|
| **Step 1: Entity Extraction Prompt** 
 Extract the entities from the following evidence that directly influence the judgment of the claim. 

 **Example:** 
 Claim: On April 6 2025: Red Sea crisis March–April 2025 United States attacks in Yemen. Between four and 70 people are killed and at least 16 others are injured in overnight U.S. airstrikes targeting Houthi forces in Saada, Yemen. 
 Evidence: ... In March 2025, the United States launched a large campaign of air and naval strikes against Houthi targets in Yemen. Code- named Operation Rough Rider, it has been the largest U.S. military operation in the Middle East of President Donald Trump's second term ... 
 Entities: ...March 2025, Yemen, Rough Rider, Donald Trump's second term... 

 **Task:** 
 Claim: {claim} 
 Evidence: {evidence} |
| **Step 2: Replaced Entity Generation Prompt** 
 Given the entity, generate a similar type but a different entity. Only output the new entity. 

 **Example:** 
 Entity: ...March 2025, Yemen, Rough Rider, Donald Trump's second term... 
 New Entity: ...March 2026, Oman, Iron Stallio, Joe Biden's second term 

 **Task:** 
 Entity: {ent} |

Table 18: Prompts of counterfactual evidence generation framework

| Counterfactual Evidence Generation |
|---|
| **True Claim Refutation Prompt** 
 Given a claim labeled as true, generate realistic but fabricated evidence to refute it. Create convincing fake content that contradicts the true claim. 

 **Task:** 
 Claim: {claim} (label: true) 
 Evidence: |
| **False Claim Support Prompt** 
 Given a claim labeled as false, generate realistic but fabricated evidence to support it. Create convincing fake content that validates the false claim. 

 **Task:** 
 Claim: {claim} (label: false) 
 Evidence: |

Table 19: The prompt structures of ImplicitSCR and ExplicitSCR

| ImplicitSCR |
|---|
| **Prompt:** You will be given a question and a evidence. The evidence may not be trustworthy. Use your judgment to assess the reliability of the evidence. Then, based on both your assessment and your own knowledge, provide the best possible answer.
**Question:** [question]
**Evidence:** [evidence]
**Answer:** |

| ExplicitSCR |
|---|
| **Prompt:** Task Overview: You will be given a question along with your internal answer, evidence that may contain either true or false information, and the evidence's answer to the same question. Your task is to evaluate the reliability of the evidence and determine whether the evidence is deceptive or not.
Steps:
1. Internal Reasoning: Reflect on how you arrived at your internal answer using your own knowledge. Break down your reasoning process and assess the confidence level of your original answer, explaining why you believe your answer is correct.
2. Evidence Evaluation: Analyze the evidence and cross-reference the information provided with the known facts you used to form your internal answer. Determine whether the evidence contains deceptive or unreliable information, considering possible contradictions or inconsistencies.
3. Final Judgment: Based on your analysis, decide which answer (your internal answer or the evidence's answer) is more likely to be correct. Clearly state your final answer.
**Question:** {question}
**Your answer:** {internal answer}
**The evidence to judge:** {evidence}
**The evidence answer:** {evidence answer}
Please provide a detailed reasoning process, followed by your final judgment. Ensure the last line of your response contains only the final answer without any additional explanation or details. |

Table 20: The prompt structures of InternalEval and ContextEval

| InternalEval |
|---|
| **Prompt:** Your task is to evaluate the model's response to a question. You will be provided with a question, the model's answer. Your job is to determine whether the model's answer is true or false.
**Question:** [question]
**Model Answer:** [model answer]
**Is the model's answer true or false?**
Return "True" if the model's answer is correct, and "False" if the model's answer is incorrect. |

| ContextEval |
|---|
| **Prompt:** You will be given a question and evidence that answers the question. Your task is to evaluate whether the evidence provides a correct answer to the question. If the evidence's answer is correct, return "True"; otherwise, return "False".
**Question:** [question]
**Evidence:** [doc]
**Is the evidence correct?**
Return "True" if the evidence's answer is correct, and "False" if the evidence's answer is incorrect. |

Table 21: The prompt structure of TACS-LR

| TACS-LR |
| --- |
| **Step 1: Evidence Filtering Prompt**
You will be given an Evidence and a question. You need to remove the sentence that you think is not correct. You can only do removal and you can not add any new information or change the existing information. Only return the filtered evidence as your output.

**Example:**
Evidence: The Eiffel Tower is located in Paris, France. It is the tallest structure in Paris. The Eiffel Tower was built in the 19th century and is made of wood.
Question: Where is the Eiffel Tower located?
Filtered Evidence: The Eiffel Tower is located in Paris, France. It is the tallest structure in Paris. The Eiffel Tower was built in the 19th century.

**Task:**
Question: {question}
Evidence: {evidence} |
| **Step 2: Final Answer Generation Prompt**
Your task is to answer the "Question" based on the provided "Evidence".

**Example:**
Question: Is the claim 'The capital of France is Paris' true or false? Do not generate the process, just answer "true" or "false" only!
Evidence: Paris has been the administrative, political, and cultural capital of France since 987 AD when the Capetian dynasty established their power base there, a status that has remained uninterrupted for over a thousand years.
Answer: true

**Task:**
Question: {question}
Evidence: {filtered evidence} |

