# OpenReview forum: "Insist when Know, Caution when Not Know?  Unveiling LLMs' Fact-Checking Behavior Amidst Knowledge Conflicts"
_ICLR.cc/2026/Conference — ICLR 2026 Conference Withdrawn Submission_

### Official Review · Reviewer_8kFW · 2025-10-22

**Soundness:** 3
**Presentation:** 3
**Contribution:** 3
**Rating:** 4
**Confidence:** 4

**Summary:**

Focusing on the conflict between LLM’s internal parametric knowledge and external evidence, this paper investigates models’ preferences considering their own confidence. By composing public fact-checking datasets, the authors build a new evaluation benchmark. Besides, they also propose an entropy-based method to generate accurate responses.

**Strengths:**

- This paper provides many insightful findings that may be useful for model selection in real RAG/fact-checking systems.
- Based on the output confidence (JSD), this paper proposes a simple and effective method to generate faithful responses.
- The experimental settings are rigid and the metrics are inspiring and may be used in other knowledge conflict tasks.

**Weaknesses:**

- The experiments focusing on evaluating different series of LLMs, ignoring the algorithms’ biases, e.g. SFT/RLHF/RLVR. I understand that they need enormous training resources.
- Although the evaluated LLMs cover different series of models, there is no consistent evaluation on the model scaling property. For example, it would be better to evaluate one or two series of models with different sizes (Qwen or LLaMA).
- Although there are many takeaways, some of them are less analyzed and insightful. For example, `Not all LLMs are naturally well-suited for handling knowledge conflict scenarios`. This conclusion and corresponding analysis are basically the description of Table 3 & 4. However, there lacks of in-depth analysis on why this happens. Why some models are prone to be affected than the others? What may be the root cause?
- The experiments are conducted on the fact checking task with external evidence. Although current metrics cover IR and PR, there are no observations on the aspect of whether the answer is `support` or `refute`. Besides `wrong` and `correct` results, maybe the original answer type matters?

Except for the SFT and RL settings, I'll raise my overall ratings if the above concerns could be addressed.

**Questions:**

- Do you have additional insights on the threshold $\tau$ selection? Although it is determined by a small held-out set, different models may demonstrate different preference. Is $\tau$ relevant with
- Suggestion: knowledge conflict is an important field in trustworthy generation, especially on RAG systems, maybe future work could extend the evaluation benchmark to a more realistic scenario? Besides the conflict between LLM’s parametric knowledge and retrieved documents, there may be conflicts in the retrieved documents.
    - Pham et al., 2024, Who’s Who: Large Language Models Meet Knowledge Conflicts in Practice
    - Su et al., 2024, ConflictBank: A Benchmark for Evaluating the Influence of Knowledge Conflicts in LLM
    - Gao et al., 2025, Probing Latent Knowledge Conflict for Faithful Retrieval-Augmented Generation
    - Wang et al., 2025, Accommodate Knowledge Conflicts in Retrieval-augmented LLMs: Towards Reliable Response Generation in the Wild

---

> ### Author Response · Authors · 2025-11-21
> **Response to Reviewer 8kFW (1/4)**
>
> Thank you for your valuable feedback to help us improve our paper. We have revised our paper based on your feedback. We detail our response below and please kindly let us know if our response addresses your concerns.
>
> > **Weakness 1** -- ``_Although the evaluated LLMs cover different series of models, there is no consistent evaluation on the model scaling property. For example, it would be better to evaluate one or two series of models with different sizes (Qwen or LLaMA)"_
>
> __A1:__ We thank the reviewer for this crucial suggestion. Following this advice, we have conducted an evaluation of the scaling properties within the Llama3 and Qwen3 series, with the results presented in Tables R1, R2, and R3. Our findings reveal distinct and informative behavioral patterns across different model sizes. __We have updated the experimental results in the current version, specifically in Section 4, "Model Scaling Evaluation".__
>
> - The Margin score (the following Table R1) shows a clear positive correlation with model size across both model families. For the Qwen3 series, the Margin consistently increases from -0.585 (8B) to 0.149 (32B). Similarly, while not strictly linear, the Llama3 series shows a dramatic leap in performance at the 8B scale, reaching a Margin of 1.626. This overarching trend suggests that increasing model size may enhance its capability to resolve knowledge conflicts.
>
> - Specificlly, as shown in the following Table R2 and R3, the relatively smaller-scale models (e.g., Llama3-1B/3B) and the smaller Qwen3 models (8B/14B) demonstrate a higher tendency toward response inertia, indicated by high Persistence Ratios (PR) even when their initial answers are incorrect (i.e., Llama3-1B/3B: 97.98%/99.22%; Qwen3-8B/14B: 85.71%/84.57%). As model scale increases, the shift towards higher Persistence Rate (PR) (e.g. 69.19% of Llama3-8B; 70.25% of Qwen3-32B) after initially correct answers and increased Influence Rate (IR) after initially wrong answers indicates an improved ability to arbitrate knowledge conflicts, pointing to a more defined self-awareness of knowledge boundaries.
>
> __Table R1__: Margin Across Llama3 and Qwen3
>
> | Model | Margin $\uparrow$| Model | Margin $\uparrow$ |
> |:---|:---:|:---|:---:|
> | **1b (Llama3)** | 0.625| **8b (Qwen3)** | -0.585 |
> | **3b (Llama3)** | 0.369 |**14b (Qwen3)** | -0.300 |
> | **8b (Llama3)** | 1.626 |**32b (Qwen3)** | 0.149  |
>
>
> __Table R2__:  Performance (IR, PR, and O) when the initial response of models is Wrong, categorized by Know and Not Know
>
> | Model | IR (Know) | PR (Know) | O (Know) | IR (Not Know) | PR (Not Know) | O (Not Know) |
> |:--- |:---:|:---:|:---:|:---:|:---:|:---:|
> | **1b (Llama3)** | 2.02% | 97.98% | 0.021 | 23.26% | 76.74% | 0.303 |
> | **3b (Llama3)** | 0.78% | 99.22% | 0.008 | 53.13% | 46.88% | 1.133 |
> | **8b (Llama3)** | 70.05% | 29.95% | 2.339 | 67.00% | 33.00% | 2.030 |
> | **8b (Qwen3)** | 14.29% | 85.71% | 0.167 | 55.52% | 44.48% | 1.248 |
> | **14b (Qwen3)** | 15.43% | 84.57% | 0.182 | 59.48% | 40.52% | 1.468 |
> | **32b (Qwen3)** | 49.41% | 50.59% | 0.977 | 39.18% | 60.82% | 0.644 |
>
> __Table R3__: Performance (IR, PR, and O) when the initial response of models is Correct, categorized by Know and Not Know
>
> | Model | IR (Know) | PR (Know) | O (Know) | IR (Not Know) | PR (Not Know) | O (Not Know) |
> |:--- |:---:|:---:|:---:|:---:|:---:|:---:|
> | **1b (Llama3)** | 0.98% | 99.02% | 0.010 | 15.92% | 84.08% | 0.189 |
> | **3b (Llama3)** | 2.21% | 99.22% | 0.022 | 27.59% | 72.40% | 0.381 |
> | **8b (Llama3)** | 30.80% | 69.19% | 0.445 | 40.16% | 59.84% | 0.672 |
> | **8b (Qwen3)** | 42.22% | 57.78% | 0.731 | 64.96% | 35.04% | 1.854 |
> | **14b (Qwen3)** | 41.90% | 58.10% | 0.721 | 60.87% | 39.80% | 1.529 |
> | **32b (Qwen3)** | 29.75% | 70.25% | 0.423 | 47.37% | 52.63% | 0.900 |

---

> ### Author Response · Authors · 2025-11-21
> **Response to Reviewer 8kFW (2/4)**
>
> > **Weakness 2** -- ``Although there are many takeaways, some of them are less analyzed and insightful. For example, Not all LLMs are naturally well-suited for handling knowledge conflict scenarios. This conclusion and corresponding analysis are basically the description of Table 3 & 4. However, there lacks of in-depth analysis on why this happens. Why some models are prone to be affected than the others? What may be the root cause?"_
>
> __A2:__ We sincerely thank the reviewer for this insightful and crucial point. The question of why some models are better at handling knowledge conflicts is indeed a complex research topic. While a definitive answer is beyond the scope of our current empirical study, we try to propose several general hypotheses based on our findings and existing literature.
>
> - __Internal Priors vs. Context__. When the provided evidence (context) conflicts with the model's strong internal priors, the model may face a dilemma. In some instances, the strength of these priors, which are learned from extensive and diverse datasets, tend to override the contextual evidence. This phenomenon is supported by several studies [1,2,3]. That said, _different models exhibit varying degrees of prior strength, which explains their differing levels of acceptance when encountering conflicting information._
>
> - __The Influence of RLHF__. Second, we hypothesized that RLHF might not only enhance model capabilities but also inadvertently reinforce its "confidence" in its own responses. However, testing this hypothesis on proprietary models like Deepseek-v3 is challenging, as their base models are not publicly available. To investigate this mechanism empirically, we conducted a targeted experiment comparing DeepSeek-V2 (the base model) with DeepSeek-V2-Chat (the aligned model) on our dataset with 500 samples. The results are shown in Table R4.
>
> __Table R4__: Comparision of Persistence Rate
> | Model | Correct Initial Response | Wrong Initial Response|
> | :--- | :---: | :---: |
> | | **Know / Not-Know** | **Know / Not-Know** |
> | **DeepSeek-V2-Chat** (RLHF) | PR: **0.774** / PR: **0.520** | PR: **0.878** / PR: **0.427** |
> | **DeepSeek-V2** (Base) | PR: 0.748 / PR: 0.507 | PR: 0.845 / PR: 0.405 |
>
> We observed that DeepSeek-V2-Chat, which has undergone RLHF training, demonstrates a higher Persistence Rate (PR) in most scenarios, particularly when its initial response was incorrect (Know: 0.878 vs. 0.845). This suggests that the alignment process may increase model's tendency to adhere to its initial beliefs regardless of correctness. _This finding indicates that the varying ability of different models to handle knowledge conflicts may be influenced by their specific RLHF data and reward settings_.
>
> __We acknowledge that this experiment is preliminary and serves as a case study. We believe a more in-depth exploration into how different alignment techniques affect epistemic humility in LLMs would be a highly valuable direction for future research.__
>
> [1]Huang P, Liu Z, Yan Y, et al. Pip-kag: Mitigating knowledge conflicts in knowledge-augmented generation via parametric pruning.
>
> [2]Yuan X, Yang Z, Wang Y, et al. Discerning and Resolving Knowledge Conflicts through Adaptive Decoding with Contextual Information-Entropy Constraint. ACL, 2024
>
> [3]Shi W, Han X, Lewis M, et al. Trusting your evidence: Hallucinate less with context-aware decoding. NAACL, 2024

---

> ### Author Response · Authors · 2025-11-21
> **Response to Reviewer 8kFW (3/4)**
>
> > **Weakness 3** -- ``The experiments are conducted on the fact checking task with external evidence. Although current metrics cover IR and PR, there are no observations on the aspect of whether the answer is support or refute. Besides wrong and correct results, maybe the original answer type matters?"_
>
>
> __A3:__ We sincerely thank the reviewer for this insightful question. We further analyzed the models' original answer ("support" vs. "refute") under different conditions in the __following Table R5 and R6__. Our analysis yields several primary findings:
>
> - As shown in the Know columns under the "Correct" section, all models demonstrate a much higher proportion of Refute cases when they answer correctly based on their internal knowledge. For instance, Deepseek-v3's correct Know answers are composed of 85.05% Refute statements, and most other models show a Refute ratio well above 60%. This strongly suggests that the reliable, pre-trained knowledge of LLMs is predominantly geared towards negation. In other words, models may be more capable of confidently and correctly identifying a false statement than confirming a true one.
>
> - In the Not-Know columns for Table R5 and R6(correct and wrong answers), most Support cases consistently outnumber Refute cases (e.g., Gpt-4o-mini: 70.89% support vs  29.11% refute in Table R5; 66.38% support vs 33.62% refute in Table R6). This indicates that affirmative statements constitute the bulk of questions that models may be uncertain about and require external tools to verify.
>
>
> __Table R5__: Distribution of support/refute labels for correct initial response
>
> | Model           | Know | Not-Know              |
> |:----------------|:-------------------------------------:|:--------------------------------:|
> |                 |   **Support (%)** / **Refute (%)**    | **Support (%)** / **Refute (%)** |
> | **Deepseek-v3** |             29.94 / 70.06             |          65.00 / 35.00           |
> | **Gemini-2.5**  |             40.88 / 59.12             |          60.79 / 39.21           |
> | **Qwen3-32B**   |             36.67 / 63.33             |          65.50 / 35.00           |
> | **Gpt-4o-mini** |             32.88 / 67.12             |          70.89 / 29.11           |
> | **Llama3-8B**   |             31.47 / 68.53             |          45.35 / 54.65           |
> | **Phi-4**       |             26.39 / 73.61            |          42.05 / 57.95           |
> | **Mistral-7B**       |             14.46 / 85.54     |   43.26 / 56.74          |
>
>
> __Table R6__: Distribution of support/refute labels for wrong initial answers
>
> | Model           | Know | Not-Know  |
> |:----------------|:-----------------------------------:|:--------------------------------:|
> |                 |  **Support (%)** / **Refute (%)**   | **Support (%)** / **Refute (%)** |
> | **Deepseek-v3** |            85.05 / 14.95            |          63.69 / 36.31           |
> | **Gemini-2.5**  |            66.92 / 33.08            |          62.01 / 37.99           |
> | **Qwen3-32B**   |            64.47 / 35.53            |          56.01 / 43.99           |
> | **Gpt-4o-mini** |            79.67 / 20.33            |          66.38 / 33.62           |
> | **Llama3-8B**   |            81.18 / 18.82            |          61.82 / 38.18           |
> | **Phi-4**       |            71.30 / 28.70            |          61.39 / 38.61           |
> | **Mistral-7B**       |  80.60 / 19.40      |   62.12 / 37.88

---

> ### Author Response · Authors · 2025-11-21
> **Response to Reviewer 8kFW (4/4)**
>
> > **Question 1** --``Do you have additional insights on the threshold  selection? Although it is determined by a small held-out set, different models may demonstrate different preference. Is  relevant with"
>
> __A4:__ Thank you for your insightful question. We hypothesize that two potential aspects determine the threshold.
> - First, _Contextual Plasticity_: This describes how external evidence influences the model's output. Models that are less influenced by evidence can present low-entropy, consistent conclusions (which potentially lead to high $\mathbf{JSD}_{\text{exp}}$ and then a small $\Delta$); in this case, a lower threshold can be used, implying a slightly more lenient criterion for choosing whether to trust the external knowledge. Conversely, models that are more susceptible to influence require a higher threshold, meaning a stricter one, to avoid over-trusting evidence that has minimal impact on their behavior and may be unreliable.
> - Second, _Confidence Bias_: Overconfident models provide extreme probability, low-entropy judgments even when incorrect (lead to low $\mathbf{JSD}_{\text{par}}$  and then a small $\Delta$); for such models, a lower threshold should be set, appropriately expanding the scope for adopting evidence. Conversely, a higher threshold can be set for others who are not overconfident.
>
> > **Suggestions** ``Suggestion: knowledge conflict is an important field in trustworthy generation, especially on RAG systems, maybe future work could extend the evaluation benchmark to a more realistic scenario? Besides the conflict between LLM’s parametric knowledge and retrieved documents, there may be conflicts in the retrieved documents."
>
> __A5:__ We thank the reviewer for this valuable suggestion. We agree that extending the evaluation to other common scenarios is a promising and important direction for future work. Additionally, we have now cited all the mentioned papers in our revised manuscript Appendix B, and distinguished how they differ from our work.
>
> ---
>
> __Manuscript Revisions:__ We have implemented the following revisions and highlighted in the current manuscript:
> 1. We have integrated the model scaling evaluation results into **Section “Evaluation Results”**, adding a dedicated paragraph and corresponding Figures to present the findings.
> 2. We have included the Refute/Support ratio results in **Appendix A.2**, accompanied by a detailed analysis of these outcomes.
> 3. As suggested by the reviewer, we have expanded the discussion on retrieved document conflicts in the *Related Work* section, now included in **Appendix B** of the updated manuscript.
>
> We sincerely hope that our responses and revisions have adequately addressed the reviewer’s concerns, and we kindly request that the reviewer consider raising the evaluation score accordingly.

---

> > ### Comment · Reviewer_8kFW · 2025-11-27
> >
> > Thanks for the explanations and further experiments. I believe the response has addressed most of my concerns and I have raised my rating score accordingly.

---

> > > ### Author Response · Authors · 2025-11-27
> > >
> > > Dear Reviewer 8kFW,
> > >
> > > We are glad to see that our response addresses your concerns. Thank you again for your valuable feedback to help us enhance our paper.

---

### Official Review · Reviewer_6Am3 · 2025-10-24

**Soundness:** 1
**Presentation:** 2
**Contribution:** 2
**Rating:** 2
**Confidence:** 4

**Summary:**

This paper systematically investigates how LLMs arbitrate between their internal parametric knowledge and conflicting external evidence during fact-checking.
The authors introduced a fine-grained evaluation framework that classifies the model's internal knowledge state for a claim as KNOW / NOT-KNOW.
The key finding is that, most LLMs resist external conflicts when confident (KNOW) but are more receptive when uncertain (NOT-KNOW).
To improve fact verification in knowledge-conflict scenarios, the paper proposes a test-time algorithm that evaluates the stability and consistency of model predictions across multiple runs. This method achieved competitive fact-checking performance compared to other baselines.

**Strengths:**

1. The two datasets provided in this paper(FactConf, NewFactConf) are valuable resources that can contribute to improving fact verification systems.

2. The distinction between Know knowledge and Not-Know knowledge presents a good method for observing the internal behavior and knowledge boundary of LLMs.

**Weaknesses:**

1. I find it difficult to fully agree with the main idea of this paper. The authors claim that “models are more susceptible to influence in the NOT-KNOW condition than in the KNOW condition.” However, it is not clear how this finding can help build a better fact verification system. According to lines 106–107, the Know/Not-Know condition is not related to truthfulness. Suppose a user wants to verify a statement and runs the model several times to check whether it “knows” or “does not know” that statement. What should the user do next? Since the user does not know the ground truth, they cannot tell whether the model’s confidence means a correct or incorrect answer. The paper should explain more clearly how this finding can help improve the accuracy of fact verification systems.

2. The explanation of the experiments is not detailed enough, which makes it hard to understand. I have several questions about the experimental design and results, which I will list in the Question section.

3. I am not sure whether the “Conflicting Evidence” examples are truly conflicting. For example, in Table 1, the evidence (e) says BTS collaborated with McDonald’s on May 26, 2021, while the constructed conflicting evidence (e−) says BTS collaborated with KFC on October 12, 2019. These two statements do not actually conflict; both can be true. For a more precise evaluation, the dataset should include examples with real knowledge conflicts, such as “pi is equal to 4.14…,” which is clearly false.

4. The readability of some tables (e.g., Table 3 and Table 5) could be improved. For example, in Table 3, the authors compare the Know/Not-Know ratio across models based on the “#” values. It would be helpful to include the actual ratio values in the table instead of just the raw numbers, so readers do not have to calculate them manually. This would make the tables easier to read and understand.

5. Some parts of the writing are too informal, such as using “vs.” in line 258 and “#” in lines 258 and 319. These should be written in a more formal way.

6. It is unclear which version of the mistral-7B model was used. On Hugging Face, there are several versions such as mistral-7B-v0.1, mistral-7B-instruct-v0.1, mistral-7B-v0.2, mistral-7B-instruct-v0.2, and so on. Since the metrics used in this paper (like Margin and Relative Odds) are sensitive to whether a case is classified as Know or Not-Know, and since performance changes depending on whether the model is instruction-tuned, the authors should clearly state which specific checkpoint was used for each model.

+) Minor typos: in Table 6, “InteralEval”-> “InternalEval,” and in line 122, “he”->“the.”

**Questions:**

1.According to line 067, the Know and Not-Know categories are determined by whether the results from M independent runs are all consistent or not. Then, in Table 3, how were the Know and Not-Know subsets determined — based on how many independent runs? If this number is the 10 runs mentioned in lines 249–251, what exactly are the values averaged in Table 3?

2. The Correct and Wrong subsets appear to be divided based on the “initial inference,” that is, the model’s first inference under a non-zero temperature compared with the ground truth. If, for example, the model’s first inference is wrong but the remaining nine inferences are correct, does this sample belong to the Correct subset or the Wrong subset?

---

> ### Author Response · Authors · 2025-11-21
> **Response to Reviewer 6Am3 (1/3)**
>
> Thank you for your constructive comments and suggestions. We have revised our paper according to your comments. We respond to your questions below and would appreciate it if you could let us know if our response addresses your concerns.
>
> > **Weakness 1** --``I find it difficult to fully agree with the main idea of this paper. The authors claim that “models are more susceptible to influence in the NOT-KNOW condition than in the KNOW condition.” However, it is not clear how this finding can help build a better fact verification system. According to lines 106–107, the Know/Not-Know condition is not related to truthfulness.
> Suppose a user wants to verify a statement and runs the model several times to check whether it “knows” or “does not know” that statement. What should the user do next? Since the user does not know the ground truth, they cannot tell whether the model’s confidence means a correct or incorrect answer. The paper should explain more clearly how this finding can help improve the accuracy of fact verification systems."
>
> __A1:__ Thank you for this insightful question. We need to clarify that __our paper's contribution is not to provide a user-facing tool for fact-checking, but to conduct a foundational behavioural analysis of existing systems that informs the very architecture of future, more intelligent fact-verification systems.__
>
> Then we explain how our two key findings can help inspire the development of better, smarter fact-verification systems:
>
> - ``Not all LLMs are naturally well-suited for handling knowledge conflicts". Some LLMs, such as Deepseek-v3, and Phi-4 frequently insist on incorrect parametric answers and refuse to revise even when provided with correct external evidence, but Gpt-4o-mini and Llama3-8B demonstrate greater robustness and are naturally better suited for such task.
>
>     __How this helps__: It gives developers a straightforward recommendation: for a reliable fact-checking tool, one is suggested to choose a model that is less stubborn when the retrieved information is of high quality.
>
> - ``LLMs are more influenced in the NOT-KNOW condition than in the KNOW condition." This is not just a theoretical observation but an exploitable characteristic.
>
>     __How this helps__: __1.__ The developers can use this to build a smarter system that dynamically adjusts its strategy. For instance, it could trigger a rigorous, multi-source verification process only when the model is in a "Not-Know" state, while trusting its parametric knowledge more in the "Know" state. This makes the verification process both more accurate and computationally efficient.  __2.__ This can also encourage developers to focus on the Know but wrong knowledge in LLMs' parametrics, which may be hard to revise by just providing correct evidence while needing more advanced methods for correction.
>
> In summary, our paper serves as a crucial diagnostic tool. It moves the discussion from "if" models can verify facts to "how" they do it, revealing both exploitable mechanisms and critical failures.

---

> ### Author Response · Authors · 2025-11-21
> **Response to Reviewer 6Am3 (2/3)**
>
> > **Weakness 2** --``I am not sure whether the “Conflicting Evidence” examples are truly conflicting. For example, in Table 1, the evidence (e) says BTS collaborated with McDonald’s on May 26, 2021, while the constructed conflicting evidence (e−) says BTS collaborated with KFC on October 12, 2019. These two statements do not actually conflict; both can be true. For a more precise evaluation, the dataset should include examples with real knowledge conflicts, such as “pi is equal to 4.14…,” which is clearly false."
>
> __A2:__ We thank the reviewer for this sharp observation. Please allow us to clarify our definition and methodology regarding "knowledge conflicts."  Following the existing works [1,2,3], we define a knowledge conflict as __a situation where the external evidence is inconsistent with the parametric knowledge, requiring arbitration to accurately evaluate a given claim.__ To be more specific, the knowledge "conflict" in our study is defined in the context of checking a specific claim. _The original, annotated evidence is designed to help the LLM make a correct judgment about the claim. In contrast, our constructed conflicting evidence is designed to introduce noise or inaccurate information that misleads the model in its judgment of that same claim._ In the example in Table 1 of our paper, while the two statements can be both true, the constructed evidence is inconsistent (and may be even not so relevant) to the knowledge required to correctly judge that claim.
>
> We agree with the reviewer that the "clearly false" evidence they mentioned (e.g., "pi is equal to 4.14...") represents __counterfactual knowledge__. While this is indeed a more strict type of knowledge conflict, we need a system to be able to handle a wide spectrum of conflicts (e.g., inconsistencies, noises) induced from external evidence, which is crucial for a realistic setting. Meanwhile, only introducing "strictly false" evidence like "pi is equal to 4.1.4..." might lead to harsh decisions on a more "soft" claim such as "pi is less than 4".  While it was not the core focus of our research, we conducted an extended experiment to discuss this specific issue in __Appendix A.1__. From this experiment, we found that counterfactual conflicts introduce high complexity for the models. Our analysis suggests that a more reasonable and effective way to handle such conflicts might be to preprocess the evidence to filter out blatant falsehoods before injecting it into the LLM.
>
> [1]Xu R, Qi Z, Guo Z, et al. Knowledge Conflicts for LLMs: A Survey. EMNLP, 2024
> [2]Accommodate Knowledge Conflicts in Retrieval-augmented LLMs: Towards Robust Response Generation in the Wild
> [3]Astute RAG: Overcoming Imperfect Retrieval Augmentation and Knowledge Conflicts for Large Language Models
>
> > **Weakness 3** --``The readability of some tables (e.g., Table 3 and Table 5) could be improved. For example, in Table 3, the authors compare the Know/Not-Know ratio across models based on the “#” values. It would be helpful to include the actual ratio values in the table instead of just the raw numbers, so readers do not have to calculate them manually. This would make the tables easier to read and understand."
>
>
> __A3:__ We thank the reviewer for this constructive feedback on improving the readability of our tables. We would like to clarify that the # column represents the number count of claims that were categorized into the "Know" or "Not-Know" categories. _Our original intention was to provide readers with a direct view of the sample size upon which the IR (Influence Rate) and PR (Parametric Rate) metrics are based._
>
> However, we agree with the reviewer that this presentation could cause unnecessary confusion. Your suggestion to include the actual ratio values is an excellent one for enhancing clarity. __We have adopted your recommendation and revised both Table 3 and 4 (in our paper) to make them easier to understand. The updated tables are now integrated into the revised manuscript.__ Thank you again for your valuable suggestion.
>
>
> > **Weakness 4** --``Some parts of the writing are too informal, such as using “vs.” in line 258 and “#” in lines 258 and 319. These should be written in a more formal way."
>
> __A4:__ Thanks, we have revised them in current version.
>
> > **Weakness 5** --``It is unclear which version of the mistral-7B model was used. On Hugging Face, there are several versions such as mistral-7B-v0.1, mistral-7B-instruct-v0.1, mistral-7B-v0.2, mistral-7B-instruct-v0.2, and so on. Since the metrics used in this paper (like Margin and Relative Odds) are sensitive to whether a case is classified as Know or Not-Know, and since performance changes depending on whether the model is instruction-tuned, the authors should clearly state which specific checkpoint was used for each model."
>
> __A5:__ Thanks for this important question. To address this, we have now clarified that we used Mistral-7B-Instruct-v0.2 in Section 3 of our paper.

---

> ### Author Response · Authors · 2025-11-21
> **Response to Reviewer 6Am3 (3/3)**
>
> > **Questions** --``1.According to line 067, the Know and Not-Know categories are determined by whether the results from M independent runs are all consistent or not. Then, in Table 3, how were the Know and Not-Know subsets determined — based on how many independent runs? If this number is the 10 runs mentioned in lines 249–251, what exactly are the values averaged in Table 3? The Correct and Wrong subsets appear to be divided based on the “initial inference,” that is, the model’s first inference under a non-zero temperature compared with the ground truth. If, for example, the model’s first inference is wrong but the remaining nine inferences are correct, does this sample belong to the Correct subset or the Wrong subset?"
>
>
> __A6:__ We apologize for the misunderstanding our previous description may have caused. We would like to clarify our method:
>
> The # values in the table are _not a simple average or based on the "initial inference", which is total counts of outcomes aggregated across all claims_. Let's take an example: for a specific claim, we perform 10 independent runs.
> - If the results are 9 correct and 1 wrong: 1. The claim is first categorized as "Not-Know." 2. Then, we add 9 to the Correct count (#) under "Not-Know" and 1 to the Wrong count (#) under "Not-Know."
> - If the results were 10 wrong runs: 1. The claim would be categorized as "Know." 2. We would then add 10 to the Wrong count (#) under "Know."
>
> To avoid any further misunderstanding, __we have adopted your suggestion and replaced the raw counts with ratios in Tables 3 and 4.__ This change more intuitively shows the probability of a model exhibiting "Know" or "Not-Know" behavior. Thank you for the helpful advice.
>
> ---
>
> __Manuscript Revisions:__ We have implemented the following revisions and highlighted them in the current manuscript:
> 1. We have converted the raw counts in **Tables 3 and 4** into ratios.
> 2. We have revised the writing format as suggested by the reviewer (removing the use of "#" and "vs.").
> 3. We have added detailed descriptions of the LLMs used, including Mistral-7B-Instruct-v0.2, in **Section "Experimental Setup (LLM Basis)"**.
>
> We sincerely hope that our responses and revisions have addressed the reviewer’s concerns, and we hope that the reviewer consider raising the evaluation scores accordingly.

---

### Official Review · Reviewer_mXv1 · 2025-10-25

**Soundness:** 3
**Presentation:** 2
**Contribution:** 2
**Rating:** 4
**Confidence:** 4

**Summary:**

This paper explores how LLMs handle conflicts between their internal parametric knowledge and external evidence in fact-checking tasks. The authors introduce a categorization of claims called "KNOW" (consistent responses across multiple independent runs, indicating confidence) or "NOT-KNOW" (inconsistent responses, indicating uncertainty). They construct two datasets: FactConf, derived from existing benchmarks by injecting conflicting evidence via entity substitution, and NewFactConf, focusing on new events likely not in the existing training data. The evaluation results on the datasets reveal that models are more receptive to external evidence in NOT-KNOW states but vary in robustness (e.g., GPT-4o-mini and Llama3-8B perform better). To improve arbitration, they propose a test-time algorithm using Jensen-Shannon Divergence (JSD) over sampled prediction probabilities to balance parametric and external knowledge, showing competitive performance against baselines.

**Strengths:**

- The paper addresses a timely and relevant issue in LLM reliability, particularly in retrieval-augmented fact-checking, where knowledge conflicts can lead to factual errors. The empirical findings provide useful insights into model-specific behaviors, such as overconfidence in erroneous parametric knowledge (e.g., in Deepseek-v3) versus balanced resistance (e.g., in GPT-4o-mini).
- The 2 newly introduced datasets, FactConf and NewFactConf, add value by enabling controlled evaluation of conflicts, including temporal ones beyond training cutoffs. NewFactConf, in particular, tests real-world scenarios with recent events, which is a practical extension.
- JSD-based arbitration is systematic, and demonstrates strong empirical results across benchmarks.

**Weaknesses:**

- Novelty is somewhat limited by existing literature. A comprehensive survey on knowledge conflicts in LLMs (arXiv:2403.08319v1) already categorizes conflicts (e.g., context-memory similar to parametric-external) and discusses fact-checking implications, uncertainty via consistency (e.g., semantic entropy, arXiv:2302.09664), and resolution strategies (e.g., disentangling sources, contrastive decoding). The KNOW/NOT-KNOW distinction echoes prior consistency-based uncertainty estimation (e.g., rephrasing or multiple generations in the survey). Additionally, ConflictBank (arXiv:2408.12076), a benchmark for evaluating LLM knowledge conflicts across misinformation and factual evolution, closely resembles FactConf/NewFactConf in scope and methodology (e.g., controlled conflict injection, model family evaluations). The paper does not cite or compare to these, which overshadows its claim of being the "first fine-grained evaluation framework".
- The entity substitution method for creating conflicts in the new datasets may not fully capture realistic misinformation, as it assumes simple replacements create plausible contradictions. Prior work (e.g., Longpre et al., 2021, cited in the paper) notes limitations in authenticity, and the survey highlights that artificial setups limit generalizability compared to real-world RAG conflicts.
- Computational overhead is a concern - while many works claim propose consistency-inspired method to improve LLM judgement reliability, the bottleneck in practical deployment of these methods is that they often require a large number of parallel generations. The approach in this paper also relies on multiple inference runs (up to 40 in analyses) for categorization and JSD computation, which could be impractical for deployment, especially for larger models. No discussion of efficiency optimizations or approximations is provided.
- The evaluation focuses on binary (True/False) fact-checking, potentially limiting insights to more nuanced tasks (e.g., multi-label or explanatory verification) and setups (e.g. factuality issues in long-form generation).

**Questions:**

- The paper evaluates seven specific LLMs—why these models? Were open-source vs. closed-source differences (e.g., access to probabilities) accounted for in JSD approximations?
- In NewFactConf, how was "new" knowledge verified beyond release dates (e.g., accounting for potential data contamination or generalization from related events)?
- Could you provide more details on the JSD computation in practice, including how vote entropy approximates probabilities for models without logit access, and its sensitivity to temperature settings?

---

> ### Author Response · Authors · 2025-11-21
> **Response to Reviewer mXv1 (1/3)**
>
> Thank you for reviewing our paper and for your valuable feedback. Below, we address your concerns point by point and we’ve revised our paper according to your suggestions. We would appreciate it if you could let us know whether your concerns are addressed by our response.
>
> > **Weakness 1** --``Novelty is somewhat limited by existing literature. A comprehensive survey on knowledge conflicts in LLMs (arXiv:2403.08319v1) already categorizes conflicts (e.g., context-memory similar to parametric-external) and discusses fact-checking implications, uncertainty via consistency (e.g., semantic entropy, arXiv:2302.09664), and resolution strategies (e.g., disentangling sources, contrastive decoding). The KNOW/NOT-KNOW distinction echoes prior consistency-based uncertainty estimation (e.g., rephrasing or multiple generations in the survey). Additionally, ConflictBank (arXiv:2408.12076), a benchmark for evaluating LLM knowledge conflicts across misinformation and factual evolution, closely resembles FactConf/NewFactConf in scope and methodology (e.g., controlled conflict injection, model family evaluations). The paper does not cite or compare to these, which overshadows its claim of being the "first fine-grained evaluation framework"
>
> __A1:__ Thank you for raising these important references. We clarify that our work establishes a unique contribution by focusing on a different dimension of evaluation:
>
> - The survey (arXiv:2403.08319v1) offers a valuable taxonomy of knowledge conflicts (intra-memory conflicts, inter-context conflicts, and context-memory conflicts), we delve deeper into context–memory conflicts. Unlike the survey, we introduce a dedicated testbed to benchmark models on these conflicts and to examine how internal confidence relates to conflict resolution—goals and contributions that are entirely distinct from the survey’s scope.
>
> - ConflictBank (arXiv:2408.12076) is a benchmark primarily focused on conflicts arising from intra-memory and inter-context conflicts. However, our paper targets the context-memory conflict specifically. ConflictBank is organized around the causes of conflicts, such as misinformation, but our evaluation framework is structured around the internal knowledge state of the model (Know vs. Not-Know). This approach allows us to directly assess whether LLMs possess a clear self-awareness of their knowledge boundaries, which is a crucial factor for reliable fact-checking that previous benchmarks do not address.
>
> - Works like Semantic Entropy (arXiv:2302.09664) aim to improve uncertainty estimation methods, which is distinct from our evaluation goal, and also different from our proposed JSD method. Specifically, our method differs fundamentally in the goal and approach of Semantic Entropy. Semantic Uncertainty relies on semantic entropy to gauge confidence from output meanings, but our method centers on output stability. We specifically introduce the stability difference ($\Delta$) before and after evidence is provided, using it with a calibrated threshold to perform arbitration between parametric knowledge and external evidence.
>
> In summary, our contribution is novel: we introduce the first fine-grained evaluation framework specifically designed to probe the LLM's knowledge boundary in knowledge conflicts, revealing that this boundary is highly related to the failure in conflict scenarios, a finding not highlighted by previous work. However, __we acknowledge their relevance and have cited them in the new revision (in Section 2 and Apeendix B)__.
>
>
>
>
> > **Weakness 2** --``The entity substitution method for creating conflicts in the new datasets may not fully capture realistic misinformation, as it assumes simple replacements create plausible contradictions. Prior work (e.g., Longpre et al., 2021, cited in the paper) notes limitations in authenticity, and the survey highlights that artificial setups limit generalizability compared to real-world RAG conflicts."
>
> __A2:__ Thank you for your valuable feedback. We acknowledge that the entity substitution method, on its surface, may appear to be a relatively simple operational approach. Our goal was to create a scalable benchmark, and we carefully curated the substitution of at least five key entities to ensure the task's difficulty. Crucially, our experimental results demonstrate that the conflicting evidence generated through this method remains challenging for current models, as shown in Table 3 and 4 of our paper.
>
> Additionally, we explored another construction of counterfactual conflicts, as you suggested. We have included the results of this experiment in __Appendix A.1__ to demonstrate its feasibility and provide a comparison.

---

> ### Author Response · Authors · 2025-11-21
> **Response to Reviewer mXv1 (2/3)**
>
> > **Weakness 3** --``Computational overhead is a concern - while many works claim propose consistency-inspired method to improve LLM judgement reliability, the bottleneck in practical deployment of these methods is that they often require a large number of parallel generations. The approach in this paper also relies on multiple inference runs (up to 40 in analyses) for categorization and JSD computation, which could be impractical for deployment, especially for larger models. No discussion of efficiency optimizations or approximations is provided."
>
> __A3:__ Thank you for your insightful feedback regarding computational overhead.
>
> First, we wish to clarify that the high number of runs (up to 40) was used exclusively for the evaluation to analyze the impact of the number of runs (`K`). __Our method's primary results are based on a much smaller and more practical number, `K=5`.__
>
> To formally address your concern, we have added a new analysis on the trade-off between performance and efficiency. As shown in the following Table R2, performance gains become marginal as `K` increases, indicating that a small number of runs is sufficient. We selected `K=5` as an optimal balance between accuracy and overhead. Crucially, our method remains highly competitive even at `K=3`, which incurs the same overhead as the InternalEval baseline (as shown in Table R1). At `K=3`, our approach with Llama3-8B achieves a WE score of 56.20 (surpassing the next-best baseline's 54.92 in our paper's Table 5) and a CE score of 63.20 (surpassing the next-best baseline's 61.17 in our paper's Table 5).
>
> These results demonstrate that our method offers a practical and effective solution, delivering strong performance even under constrained computational budgets. __We have incorporated this analysis into the revised manuscript in Section 5.__
>
> **Table R1**: Computational Overhead of baselines and our method (Number of Inferences per Claim)
> | Method | Overhead |
> |:---|:---:|
> | ImplicitSCR | 2 |
> | ExplicitSCR | 2 |
> | InternalEval | 3 |
> | ContextEval | 2 |
> | ContextConf | 2 |
> | InternalConf | 2 |
> | TPC | 2 |
> | TACS-LR | 2 |
> | **Ours (K=3)** | **3** |
> | **Ours (K=5)** | **5** |
>
> **Table R2**: Performance of GPT-4o and Llama3-8B within Number of Runs (`K`)
> | K (Independent Runs) | WE % (GPT-4o) | CE % (GPT-4o) | OA % (GPT-4o) | WE % (Llama3-8B) | CE % (Llama3-8B) | OA % (Llama3-8B) |
> |:---:|:---:|:---:|:---:|:---:|:---:|:---:|
> | 3 | 54.80 | 65.12 | 59.96 | 56.20 | 63.20 | 59.70 |
> | 5 | 55.38 | 67.14 | 61.26 | 57.20 | 64.80 | 61.00 |
> | 10 | 56.02 | 67.14 | 61.58 | 59.00 | 65.20 | 62.10 |
> | 15 | 56.02 | 65.53 | 60.78 | 58.00 | 65.60 | 61.80 |
> | 20 | 58.00 | 65.60 | 61.80 | 58.20 | 63.80 | 61.00 |
>
>
>
> > **Weakness 4** --``The evaluation focuses on binary (True/False) fact-checking, potentially limiting insights to more nuanced tasks (e.g., multi-label or explanatory verification) and setups (e.g., factuality issues in long-form generation)."
>
> __A4:__ Thank you for this suggestion. While our current approach is not designed for generating free-form explanations, it is indeed applicable to multi-label classification tasks. To validate this, we conducted new experiments (3000 claims) on the PubHealth dataset[1], which involves classifying claims into multiple categories (e.g., "True," "False," "Unproven"). The results are shown in the following Tables R3 and R4.
>
> We observe that when applied to the multi-label scenario, the margin for Phi-4 remains negative, while for Llama3-8B, it remains positive. This finding is consistent with their respective performances on the binary-label task, as presented in our main manuscript.
>
> **Table R3**: Detailed Performance when Initial Response Correct
>
> | Model | IR (Know) | PR (Know) | O (Know) | IR (Not-Know) | PR (Not-Know) | O (Not-Know) |Margin $\uparrow$|
> |:---|:---:|:---:|:---:|:---:|:---:|:---:|:---:|
> | **Phi-4** | 29.50% | 70.49% | 0.418 | 54.60% | 45.39% | 1.203 |-0.08 |
> | **Llama3-8B**| 30.71% | 69.29% | 0.443 | 39.62% | 60.38% | 0.656 | 0.0325 |
>
> **Table R4**: Detailed Performance when Initial Response Wrong
> | Model | IR (Know) | PR (Know) | O (Know) | IR (Not-Know) | PR (Not-Know) | O (Not-Know)
> |:---|:---:|:---:|:---:|:---:|:---:|:---:|
> | **Phi-4** | 39.47% | 60.53% | 0.652 | 44.71% | 55.29% | 0.809 |
> | **Llama3-8B**|26.41% | 73.59% | 0.359 | 44.61% | 55.39% | 0.805 |
>
> [1]Kotonya N, Toni F. Explainable automated fact-checking for public health claims.
>
> > **Question 1** --``The paper evaluates seven specific LLMs—why these models?"
>
> __A5:__ Thanks for this question, and we have highlighted them in current version. We conduct evaluation on seven leading LLMs, including both closed-source (GPT-4o-mini, Gemini-2.5) and open-source models (Mistral-7B (Mistral-7B-Instruct-v0.2), Llama3-8B, Phi4, Qwen3-32B, Deepseek-v3), spanning a wide range of scales (from 7B, 8B, 14B, 32B to 617B parameters). This diversity allows for robust comparisons across different architectures, capabilities, and development paradigms.

---

> ### Author Response · Authors · 2025-11-21
> **Response to Reviewer mXv1 (3/3)**
>
> > **Question 2** --1. ``Were open-source vs. closed-source differences (e.g., access to probabilities) accounted for in JSD approximations?  2.Could you provide more details on the JSD computation in practice, including how vote entropy approximates probabilities for models without logit access, and its sensitivity to temperature settings?"
>
> __A6:__ Thanks for your insightful questions.
>
> 1) Our methodology employs two distinct approximations of the JSD to ensure a fair and robust stability evaluation across all model types:
> - For open-source models where full probability distributions (logits) are accessible, we compute the standard JSD using the soft labels. This involves calculating the entropy of the average probability distribution across K runs and subtracting the average entropy of the individual distributions.
> - For closed-source (or API-based) models where only hard labels (final decisions) are available, we approximate the JSD using vote entropy. This frequency-based method estimates an empirical probability from the K hard labels $(\hat{p} = k_{\text{true}} / K$), where $k_{\text{true}} \in [0, K]$ denotes the number of labels that belong to any of the true categories. And computes its binary entropy ($H_{\text{binary}}(\hat{p}) = -\hat{p} \log_2(\hat{p}) - (1 - \hat{p}) \log_2(1 - \hat{p})$), providing a comparable measure of output stability without requiring internal probability data. The details in __Appendix C.2 "The Voted-based JSD"__.
>
> 2) We conduct experiments for testing the __sensitivity to temperature settings__. The results shown in the following Tables R5 and R6 within wrong evidence (WE), correct evidence (CE), and the overall accuracy (OA). Regarding the impact of temperature, experimental results on Llama3-8B and Gpt-4o-mini show that increasing the temperature from 0.1 to 0.5 leads to noticeable performance gains across WE, CE, and OA. However, no significant or consistent improvement is observed when the temperature rises from 0.5 to 1.0.
>
> __Table R5__: Performance of GPT-4o across different temperatures
>
> | Temperature (t) | WE (%) | CE (%) | OA (%) |
> | :-------------: | :----: | :----: | :----: |
> |       0.1       |  51.2  |  66.0  | 58.60  |
> |       0.3       |  51.0  |  66.2  | 58.60  |
> |       0.5       |  54.0  |  68.0  | 61.00  |
> |       0.7       |  54.0  |  67.1  | 60.55  |
> |       1.0       |  55.4  |  67.1  | 61.25  |
>
> __Table R6__: Performance of Llama3-8B across different temperatures
>
> | Temperature (t) | WE (%) | CE (%) | OA (%) |
> | :-------------: | :----: | :----: | :----: |
> |       0.1       |  57.6  |  64.6  |  61.1  |
> |       0.3       |  57.6  |  64.6  |  61.1  |
> |       0.5       |  59.2  |  65.0  |  62.1  |
> |       0.7       |  58.2  |  65.6  |  61.9  |
> |       1.0       |  59.0  |  65.2  |  62.1  |
>
>
>
>
> > **Question 3** --``In NewFactConf, how was "new" knowledge verified beyond release dates (e.g., accounting for potential data contamination or generalization from related events)?"
>
> __A7:__ Thank you for this insightful question regarding how the model generates correct statements about "new" events. Our analysis suggests two main possibilities:
>
> - Powerful generalization and reasoning: The model might leverage its knowledge base to make reasonable inferences. For example, it could predict details of a recurring event based on historical patterns, demonstrating valuable reasoning.
>
> - Unverifiable hallucination: In other cases, the model may produce detailed predictions that lack authoritative backing, which we classify as a form of hallucination.
>
> We provide a concrete analysis in Appendix A.5, illustrating a scenario where the model makes an overconfident extrapolation beyond its verified knowledge. Although not outright fabrication, this can mislead users into treating speculation as fact.
>
> We acknowledge that a deeper understanding of these mechanisms is a crucial objective for ongoing research.
>
> ---
>
> __Manuscript Revisions:__ We have implemented the following revisions and highlighted them in the current manuscript:
> 1. We have integrated the evaluation results of different temperature settings and the efficiency evaluation of methods in __Section 5, "Method and Results (Analysis of efficiency and temperature)"__. And add the explicit overhead of all the baselines in __Table 5__.
> 2. We have added detailed descriptions of the LLMs used in **Section "Experimental Setup (LLM Basis)"**..
> 3. As suggested, we have cited the relevant paper in the new revision **Section 2 and Appendix B)** accordingly.
>
> We sincerely hope that our responses and revisions have adequately addressed the reviewer’s concerns, and we hope that the reviewer considers raising the score accordingly.

---

### Official Review · Reviewer_dxDn · 2025-11-01

**Soundness:** 3
**Presentation:** 3
**Contribution:** 3
**Rating:** 4
**Confidence:** 4

**Summary:**

This paper looks at a model's output under knowledge conflict. When it would output something but the context, or a rag tool, provides a conflicting example. They separate two settings, stemming from a bimodal distribution of confidence, one where the model is highly confident which they call KNOW and one where it's not which they call NOT KNOW. they create a dataset of knowledge conflicts for each model, and test their reaction, then propose a method to better steer the choice between parametric and external knowledge.

**Strengths:**

Good to show effect of some variations and ablations - run count and temperature. Good reproducibility, including on prompts. Relevant to the topic, good use and analysis of the different models, which are not just brought as quantitative empirical evidence, the differences between models are analysed and discussed.

**Weaknesses:**

I find the following work relevant to your topic, and should probably be cited:
Ortu, Francesco, et al. "Competition of Mechanisms: Tracing How Language Models Handle Facts and Counterfactuals." Proceedings of the 62nd Annual Meeting of the Association for Computational Linguistics (Volume 1: Long Papers). 2024

As is usual in this field, the use of the word "knowledge" is confusing. Here it is taken to mean high confidence, though sometimes implies correctness. This might bring more confusion than clarity.

Your method looks at multiple runs, instead of making conclusions from a single forward pass. One strong reason other methods avoid it is to not pay the extra compute. Maybe this should also be measured next to accuracy for each method.
See questions.

**Questions:**

Do I understand correctly that each model gets it's own dataset ? The type of conflicting evidence probably has a strong impact on the rejection rate here I understand you are trying to emulate likely/plausible conflicting evidence.
--> does this seem to be working from LLM sampling? Was it tested or just accepted
--> the LLM self injects variations. Could there be a difference in behaviour depending on high confidence/low confidence?

The knowledge evaluation method by sampling true/false is entirely prompt based. We know those to have a lot of variance depending on prompt (input) and chosen output tokens to compare during constrained decoding.
for example here:

Mahaut, M., Aina, L., Czarnowska, P., Hardalov, M., Mueller, T., & Màrquez, L. (2024, August). Factual Confidence of LLMs: on Reliability and Robustness of Current Estimators. In Proceedings of the 62nd Annual Meeting of the Association for Computational Linguistics (Volume 1: Long Papers) (pp. 4554-4570).

Have you checked for variation in the input, and variation in the expected output tokens (true, True, _true, Yes, ...) ?
I wonder about this in the mistral and Phi settings where they seem to be unstable and refuse external output. Could this be due to another token being most likely and constrained deciding working on the tail of the distribution?

---

> ### Author Response · Authors · 2025-11-21
> **Response to Reviewer dxDn (1/3)**
>
> Thank you for reviewing our manuscript and for your thoughtful feedback. Below, we respond to each point in turn and have revised the paper accordingly. We would appreciate it if you could let us know whether your concerns are addressed by our response.
> > **Weakness 1** --``I find the following work relevant to your topic, and should probably be cited: Ortu, Francesco, et al. "Competition of Mechanisms: Tracing How Language Models Handle Facts and Counterfactuals." Proceedings of the 62nd Annual Meeting of the Association for Computational Linguistics (Volume 1: Long Papers). 2024"
>
> __A1:__ Thank you for the suggestion. We have added the recommended work to our "Related Work" section as advised.
>
> > **Weakness 2** --``As is usual in this field, the use of the word "knowledge" is confusing.  Here it is taken to mean high confidence, though sometimes implies correctness. This might bring more confusion than clarity."
>
> __A2:__ Thank you for raising this important point. We agree that clarifying the term "knowledge" is essential.
> - In our paper, we define "knowledge" in a strictly __neutral__ sense, referring to information stored in either parametric or external sources. It is crucial to emphasize that, in our context, "knowledge" is explicitly NOT equated with high model confidence or factual correctness. _Our definitions of "Know" and "Not-Know" measure the model's certainty in its knowledge, irrespective of whether that knowledge is correct. In other words, a scenario where the model is highly certain about incorrect information/knowledge is entirely possible._
> - Our usage is consistent with established conventions in the fields of Knowledge Conflicts and Model/Knowledge Editing. _These research areas are fundamentally concerned with instances where a model's internal or external knowledge is "incorrect" and needs to be addressed or edited_. Therefore, a definition that allows for "incorrect knowledge" is a prerequisite for the entire line of inquiry [1, 2, 3, 4].
>
> To prevent any ambiguity, we have revised the manuscript to explicitly state our definition in the paper in __footnote 1__ on page 2.
>
> [1] IRCAN: Mitigating Knowledge Conflicts in LLM Generation via Identifying and Reweighting Context-Aware Neurons.
>
> [2] Wang M, Yao Y, Xu Z, et al. Knowledge mechanisms in large language models: A survey and perspective. EMNLP, 2024
>
> [3] Zhang N, Yao Y, Tian B, et al. A comprehensive study of knowledge editing for large language models.
>
> [4] Xu R, Qi Z, Guo Z, et al. Knowledge Conflicts for LLMs: A Survey. EMNLP, 2024
>
>
> > **Weakness 3** --``Your method looks at multiple runs, instead of making conclusions from a single forward pass. One strong reason other methods avoid it is to not pay the extra compute. Maybe this should also be measured next to accuracy for each method. See questions."
>
> __A3:__ Thanks for your helpful suggestions, we provide the overhead analysis as following Table R1, and we have cooperated it into the manifest.
>
> **Table R1**: The overhead across baselines and our methods
> | Method | Overhead |
> |:---|:---:|
> | ImplicitSCR | 2 |
> | ExplicitSCR | 2 |
> | InternalEval | 3 |
> | ContextEval | 2 |
> | ContextConf | 2 |
> | InternalConf | 2 |
> | TPC | 2 |
> | TACS-LR | 2 |
> | **Ours (K=3)** | **3** |
> | **Ours (K=5)** | **5** |
>
> Addtionally, we have added a new analysis on the trade-off between performance and efficiency. As shown in the following Table R2, performance gains become marginal as `K` increases, indicating that a small number of runs is sufficient. We selected `K=5` as an optimal balance between accuracy and overhead. Crucially, our method remains highly competitive even at `K=3`, which incurs the same overhead as the InternalEval baseline. At `K=3`, our approach with Llama3-8B achieves a WE score of 56.20 (surpassing the next-best baseline's 54.92 in our paper's Table 5)) and a CE score of 63.20 (surpassing the next-best 61.17 in our paper's Table 5)).
>
> These results demonstrate that our method offers a practical and effective solution, delivering strong performance even under constrained computational budgets. __We have incorporated this analysis into the revised manuscript in Section 5.__
>
> **Table R2**: Performance of GPT-4o and Llama3-8B within Number of Runs (`K`)
> | K (Independent Runs) | WE (GPT-4o) | CE (GPT-4o) | OA (GPT-4o) | WE (Llama3-8B) | CE (Llama3-8B) | OA (Llama3-8B) |
> |:---:|:---:|:---:|:---:|:---:|:---:|:---:|
> | 3 | 54.80 | 65.12 | 59.96 | 56.20 | 63.20 | 59.70 |
> | 5 | 55.38 | 67.14 | 61.26 | 57.20 | 64.80 | 61.00 |
> | 10 | 56.02 | 67.14 | 61.58 | 59.00 | 65.20 | 62.10 |
> | 15 | 56.02 | 65.53 | 60.78 | 58.00 | 65.60 | 61.80 |
> | 20 | 58.00 | 65.60 | 61.80 | 58.20 | 63.80 | 61.00 |

---

> ### Author Response · Authors · 2025-11-21
> **Response to Reviewer dxDn (2/3)**
>
> > **Question 1** --``Do I understand correctly that each model gets it's own dataset ?
> The type of conflicting evidence probably has a strong impact on the rejection rate here I understand you are trying to emulate likely/plausible conflicting evidence.--> dos this seem to be working from LLM sampling? Was it tested or just accepted --> the LLM self injects variations. Could there be a difference in behaviour depending on high confidence/low confidence?"
>
> __A4:__ We thank the reviewer for these important questions, which allow us to clarify our experimental methodology and dataset. We will address each point in order.
>
> - On the dataset used for each model: To ensure a fair and direct comparison, all models were evaluated on the exact same dataset. We understand the confusion may have arisen from our description of how the NewFactConf dataset was structured around model release dates. While it is true that we crawled different subsets of data based on each model's knowledge cut-off date (e.g., earlier models have more potential "new facts"), for the comparative evaluation presented in the paper, we used the intersection of these datasets—specifically, the smallest common set applicable to all models. We have revised the "Experimental Results" section to make this point more explicit and avoid any ambiguity.
>
> - On the plausibility of conflicting evidence: We implemented a multi-stage quality control process to ensure the plausibility and reliability of the generated conflicting evidence. This was not merely accepted but was systematically filtered and validated: During generation, we enforced several constraints, such as ensuring the substituted entity was of the same type as the original and that at least five key entities could be extracted from the context to ensure sufficient complexity. After the automated generation, we conducted a manual sampling review of the data by three humans. This process ensures the quality and challenge of our dataset, which is shown in the Appendix.
>
> We hope this response can address your concerns. If we have misinterpreted any part of your question, we would be grateful for the opportunity to clarify further.

---

> ### Author Response · Authors · 2025-11-21
> **Response to Reviewer dxDn (3/3)**
>
> > **Question 2** --``The knowledge evaluation method by sampling true/false is entirely prompt based. We know those to have a lot of variance depending on prompt (input) and chosen output tokens to compare during constrained decoding. for example here:
> Mahaut, M., Aina, L., Czarnowska, P., Hardalov, M., Mueller, T., & Màrquez, L. (2024, August). Factual Confidence of LLMs: on Reliability and Robustness of Current Estimators. In Proceedings of the 62nd Annual Meeting of the Association for Computational Linguistics (Volume 1: Long Papers) (pp. 4554-4570).
> Have you checked for variation in the input, and variation in the expected output tokens (true, True, _true, Yes, ...) ? I wonder about this in the mistral and Phi settings where they seem to be unstable and refuse external output. Could this be due to another token being most likely and constrained deciding working on the tail of the distribution?"
>
>
> __A5:__ We thank the reviewer for this insightful question about the model's sensitivity to output token formats. To rigorously test this, we conducted a new experiment on Mistral-7B using a 500-sample subset of our data. We designed __10 prompts by varying the output constraints across 5 pairs—accurant/inaccurant, true/false, correct/incorrect, support/refute, and right/wrong__—including their capitalized versions, and set the temperature to 0.5. To measure the consistency of the fact-checking results within differnent output constrains, we calculated __Fleiss' Kappa__, and the results shown in the following two tables (R3 and R4). Specifically, the results show a very high level of agreement: _the overall Fleiss' Kappa across all 10 prompts was 0.8856, indicating high agreement_. All pairwise Kappa scores were above 0.78, with many exceeding 0.95. This experiment demonstrates that the model's performance is relatively robust and not an artifact of the specific output tokens chosen.
>
> __Table R3__ : Inter-Agreement Analysis (Fleiss' Kappa)
> | Indicator | Value |
> |:---|:---|
> | Fleiss' Kappa $\uparrow$ | 0.8856 |
> | Observed Agreement (Po) | 0.9830 |
> | Expected Agreement (Pe) | 0.8517 |
> | Number of Samples | 465 |
> | Number of Raters (Prompts) | 10 |
>
> __Table R4__: Pairs Kappa Agreement Matrix
>
> |  |  **Accurant** |  **accurant** | **Correct** | **correct** | **True** | **true** | **Support** | **support** | **Right** | **right** |
> |:---|:---:|:---:|:---:|:---:|:---:|:---:|:---:|:---:|:---:|:---:|
> | **Accurant** | - | 0.8458 | 0.9173 | 1.0000 | 0.8458 | 0.8458 | 1.0000 | 0.8458 | 0.7832 | 0.7832 |
> | **accurant** | 0.8458 | - | 0.9275 | 0.8458 | 1.0000 | 1.0000 | 0.8458 | 0.8632 | 0.9353 | 0.9353 |
> | **Correct** | 0.9173 | 0.9275 | - | 0.9173 | 0.9275 | 0.9275 | 0.9173 | 0.9275 | 0.8634 | 0.8634 |
> | **correct** | 1.0000 | 0.8458 | 0.9173 | - | 0.8458 | 0.8458 | 1.0000 | 0.8458 | 0.7832 | 0.7832 |
> | **True** | 0.8458 | 1.0000 | 0.9275 | 0.8458 | - | 1.0000 | 0.8458 | 0.8632 | 0.9353 | 0.9353 |
> | **true** | 0.8458 | 1.0000 | 0.9275 | 0.8458 | 1.0000 | - | 0.8458 | 0.8632 | 0.9353 | 0.9353 |
> | **Support** | 1.0000 | 0.8458 | 0.9173 | 1.0000 | 0.8458 | 0.8458 | - | 0.8458 | 0.7832 | 0.7832 |
> | **support** | 0.8458 | 0.8632 | 0.9275 | 0.8458 | 0.8632 | 0.8632 | 0.8458 | - | 0.8058 | 0.8058 |
> | **Right** | 0.7832 | 0.9353 | 0.8634 | 0.7832 | 0.9353 | 0.9353 | 0.7832 | 0.8058 | - | 1.0000 |
> | **right** | 0.7832 | 0.9353 | 0.8634 | 0.7832 | 0.9353 | 0.9353 | 0.7832 | 0.8058 | 1.0000 | - |
>
>
> ---
> __Manuscript Revisions:__ we have _implemented the following revisions and highlighted them in the current manuscript_:
> 1. We have added the explicit overhead of all the baselines in __Table 5__. And we have integrated the evaluation results of different temperature settings and the efficiency evaluation of methods in __Section 5, "Method and Results (Analysis of efficiency and temperature)"__.
> 2. As suggested, we have cited the relevant paper in the new revision **Section 2 "Related Works"** accordingly.
> 3. We have revised the manuscript to explicitly state our definition of "knowledge" in **footnote 1** on page 2 to prevent any ambiguity.
>
> We sincerely hope that our responses and revisions have adequately addressed the reviewer’s concerns, and we hope that the reviewer consider raising the rating accordingly.

---

### Note · Authors · 2025-12-03

I have read and agree with the venue's withdrawal policy on behalf of myself and my co-authors.